# The Loss Is Not Enough: Sampling Conditions and Inductive Bias in Contrastive Representation Learning

Justinas Zaliaduonis [1 2]  Patrick Putzky [3]  Till Richter [1 4]  Sergios Gatidis [5]

## Abstract

Contrastive learning has become a leading paradigm for self-supervised representation learning, yet the conditions under which it recovers meaningful latent geometry remain incompletely understood. We develop a measure-theoretic framework formalizing the diversity condition, a support requirement on positive-pair sampling that is necessary for isometric latent recovery. We show that the standard full-support von Mises-Fisher setting implies the satisfaction of the diversity condition and as a consequence global contrastive loss minimizers recover latent geometry up to orthogonal transformation, while restricted conditionals can make non-orthogonal maps attain strictly lower asymptotic contrastive loss. We introduce a support-corrected Information Noise Contrastive Estimation (InfoNCE) variant as a theoretical fix: this correction makes orthogonal latent space recovery achievable but does not uniquely select it. Experiments on synthetic benchmarks validate the identifiability predictions, and CIFAR-10 experiments are consistent with the qualitative prediction that architectural inductive bias becomes more important when sampling diversity is limited. Together, our results clarify how sampling mechanisms and encoder inductive bias interact in contrastive representation learning.

[*]Equal contribution  [1]Technical University of Munich, Munich, Germany [2]Stanford University, Stanford, USA [3]Merantix Momentum GmbH, Berlin, Germany [4]Helmholtz Munich, Munich, Germany [5]Department of Radiology, Stanford University School of Medicine, Stanford, USA. Correspondence to: Justinas Zaliaduonis <justinas.zaliaduonis@gmail.com>, Patrick Putzky <patrick.putzky@merantix-momentum.com>, Till Richter <till.richter@helmholtz-munich.de>, Sergios Gatidis <sgatidis@stanford.edu>.

*Proceedings of the 43$^{rd}$ International Conference on Machine Learning*, Seoul, South Korea. PMLR 306, 2026. Copyright 2026 by the author(s).

## 1. Introduction

The machine learning community has long envisioned methods that turn vast amounts of unlabeled data into dense, robust, and reusable representations useful for many different downstream tasks such as classification, regression, and search. Contrastive learning (CL) has emerged as a successful technique for achieving this goal, which in recent years has led to advances in language (Jaiswal et al., 2021), vision (Chen et al., 2020), video (Zhao et al., 2024), and multimodal (Radford et al., 2021) domains.

The vast range of applications in scientific fields like Biology (Richter et al., 2025; Bahrami et al., 2025), Physics (Cy et al., 2023; Wilkinson et al., 2025), and Climate Science (Ballard, 2022; Liu et al., 2026) has made CL one of the most widely adopted unsupervised learning methods (Uelwer et al., 2023). However, despite its empirical success, the precise mechanisms driving contrastive learning remain only partially understood. This gap in theoretical understanding results in heuristic-driven development, inefficient use of computational resources, and design choices that may not fully exploit the method's potential. In this work, we seek to move beyond intuitive understanding of CL and provide a rigorous framework to reason about its regimes of success and failure.

One approach to explain the learning mechanisms posits that CL induces data representations invariant to nuisance factors (Dangovski et al., 2022; Liu et al., 2025; Poudel et al., 2022). However, this framework does not address which factors in the data are nuisance, nor how the choice of data augmentations implicitly determine this partition. Moreover, the choice of nuisance factors, often referred to as the style-content decomposition (von Kügelgen et al., 2021), can be detrimental to downstream tasks: depending on the intended use of the learned representations, factors deemed "nuisance" by the contrastive objective may carry discriminative information necessary for a specific downstream task. We refer to this approach as the ***Invariance Explanation***.

An alternative direction reasons that CL recovers the "true" generating factors of the data (Ji et al., 2023; Kirchhof et al., 2023; Sandilya et al., 2025). This approach assumes that

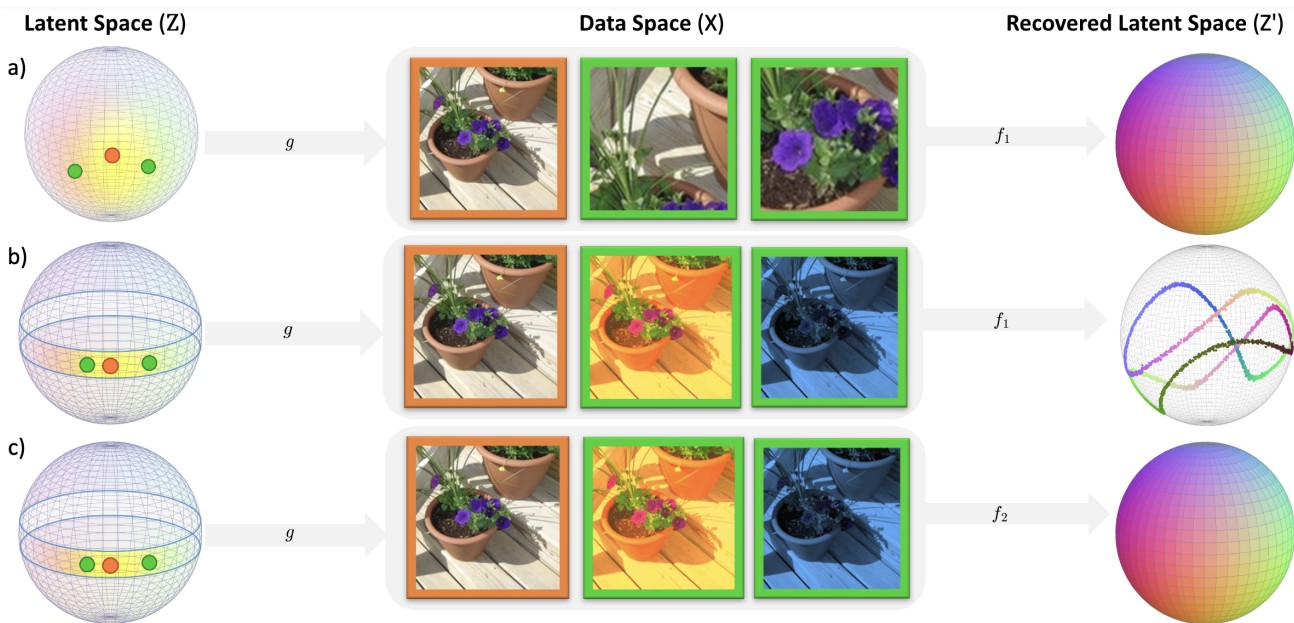

Latent Space (Z)  Data Space (X)  Recovered Latent Space (Z')

*Figure 1.* Overview of contrastive learning and the role of sampling diversity and inductive bias. The generative process $g$ maps latent variables to observations, and the encoder $f$ learns to recover the latent structure. Here $f_1$ denotes a low inductive bias encoder (e.g., MLP) and $f_2$ a high inductive bias encoder (e.g., a model of the inverse process). Orange dot indicates the anchor point; green dots are co-occurring (positive) samples. Border colors on images match their latent positions. (a) Diversity holds: $f_1$ recovers geometry. (b) Diversity violated (blue band): $f_1$ fails. (c) Diversity violated: $f_2$ recovers the latent structure despite restricted sampling diversity.

data lies on a high-dimensional manifold but possesses a low-dimensional latent structure, and that a generative process maps this low-dimensional representation to the observed high-dimensional data. In this direction, CL recovers the latent structure. Figure 1 illustrates this setup. We refer to this approach as the ***Recovery Explanation***.

In this work, we argue that the Recovery Explanation provides a more epistemologically complete account of the mechanisms underlying contrastive learning. We introduce a constraint on the conditional law of the latent space, $P_{\tilde{Z}|z}$, which we term the ***diversity condition***, and show that it is necessary for recovering the latent space up to an isometric (distance-preserving) transformation. Building on the notion of latent space recovery introduced by (Zimmermann et al., 2021), we separate the classical full-support vMF setting from the practically relevant setting where sampling is restricted. Our proof of the full-support case uses a probabilistic Mazur-Ulam argument (Zaliaduonis & Gatidis, 2026) and does not require differentiability of the recovery map. Our main result shows that violated diversity can make geometry-distorting solutions preferable, and that correcting the support mismatch restores, but does not uniquely select, geometry-preserving minimizers.

The real-world sampling mechanisms often violate the diversity condition and the latent structure is typically unknown a priori. We propose a generalized InfoNCE objec-

tive to address this gap and prove that this adjustment allows geometry-preserving latent spaces to be among the optimal solutions of the objective.

We empirically study the qualitative implications of the theory on CIFAR-10 (Krizhevsky, 2009). Here, we examine how architectural inductive bias affects representation quality under different augmentation regimes. In summary, our contributions are:

- Formalize a measure-theoretic diversity condition on latent space sampling, distinguish it from the full-support vMF assumption used in prior identifiability results, and analyze what fails when this condition is violated.

- Prove that violated diversity can make non-orthogonal recovery maps attain strictly lower asymptotic contrastive loss than any orthogonal map, and show that a support-corrected InfoNCE objective makes isometric embeddings achievable.

- Empirically validate theoretical predictions on synthetic datasets and examine their qualitative implications on CIFAR-10, demonstrating how sampling strategies and encoder inductive bias jointly determine representation quality.

- Distill design guidance for choosing augmentation

pipelines and encoder architectures from the theory-experiment alignment.

## 2. Related Work

**Identifiability in Contrastive Learning.** The question of identifiability in unsupervised learning has a long history, beginning with classical results in Independent Component Analysis (ICA). Early work established that linear mixtures of non-Gaussian independent sources can be identified up to permutation and scaling, laying the theoretical foundation for blind source separation (Comon, 1994). Subsequent developments provided practical algorithms and characterized the fundamental limits of linear ICA (Hyvärinen & Oja, 2000). The broader goal of learning disentangled representations that capture meaningful factors of variation was later articulated (Bengio et al., 2013), though it was subsequently proved that unsupervised disentanglement is impossible without inductive biases on both the model and data (Locatello et al., 2019).

In the context of contrastive learning, a formal definition of latent space identifiability was introduced, with proofs that InfoNCE recovers ground-truth factors up to orthogonal transformation (Zimmermann et al., 2021). This analysis employs von Mises-Fisher conditionals and cross-entropy asymptotics on spherical manifolds. Connections between contrastive objectives and nonlinear ICA have also been established, demonstrating identifiability through temporal structure (Hyvärinen & Morioka, 2016).

**Augmentations and Invariance.** A complementary line of work examines how data augmentations shape learned representations by defining which factors should be preserved versus discarded. A content-style framework formalizes this intuition, proving that augmentation-based contrastive learning achieves block-identifiability of the invariant content partition under a latent variable model with nontrivial statistical and causal dependencies (von Kügelgen et al., 2021). The InfoMin principle proposes that optimal views for contrastive learning should share minimal mutual information while retaining task-relevant information, thereby discarding nuisance factors (Tian et al., 2020). A causal interpretation shows that data augmentations can be viewed as interventions on style variables, motivating an explicit invariance regularizer that enforces invariant prediction of proxy targets across augmentations (Mitrovic et al., 2021). Wen & Li (2021) study a complementary feature-learning regime: they analyze gradient-based learning in finite ReLU networks and show how suitable augmentations can decouple desired sparse features from nuisance dense features. Our analysis asks a different question, namely whether the asymptotic contrastive objective recovers latent geometry up to isometry, and how architectural inductive bias compensates when sampling diversity is insufficient.

**Architectural Bias and Extensions.** The role of encoder architecture in contrastive learning has received increasing attention. Recent work analyzes how architectural constraints influence the geometry of learned representations (HaoChen & Ma, 2023), while theoretical frameworks have been extended to multimodal settings (Tschannen et al., 2023). Other contributions address gaps between theoretical assumptions and practical implementations (Rusak et al., 2025).

**Empirical Methods.** Our theoretical analysis builds upon empirically successful methods including SimCLR (Chen et al., 2020), Contrastive Predictive Coding (CPC) (van den Oord et al., 2019), and VICReg (Bardes et al., 2022). These frameworks demonstrate the practical efficacy of contrastive objectives across vision, language, and multimodal domains.

## 3. Preliminaries

### 3.1. Contrastive Learning Framework

Following (Zimmermann et al., 2021), we analyze contrastive learning using tools from Nonlinear Independent Component Analysis (ICA) (Hyvärinen et al., 2019). We consider an encoder $f : \mathcal{X} \to \mathcal{Z}$ mapping observations from the data space $\mathcal{X} \subset \mathbb{R}^n$ to a latent representation space $\mathcal{Z} \subset \mathbb{R}^k$, where $k < n$. We model the data as generated by an injective mapping $g : \mathcal{Z} \to \mathcal{X}$ from a lower-dimensional latent manifold whose coordinates represent statistically independent factors (Figure 1).

Contrastive learning aims to learn encoder parameters $\theta$ by discriminating between co-occurring signals $x, \tilde{x} \in \mathcal{X}$ (Chen et al., 2020). These signals may arise from natural mechanisms (e.g., different modalities of the same scene) or synthetic transformations (e.g., image augmentations). We define a view-generating function $\tau : \mathcal{X} \to \mathcal{X}$ that produces related views $\tilde{x} = \tau(x)$, and aim to learn an encoder such that the underlying latent factors are identified.

### 3.2. Sampling Mechanism

Since the data is generated via the injective mapping $g$, observations satisfy $x = g(z)$ and $\tilde{x} = \tau(g(z)) = g(\tilde{z})$. As view generation is typically non-deterministic, we treat it stochastically through the conditional law $P_{\tilde{X}|x}$. This formulation connects to the latent dynamics $P_{\tilde{Z}|z}$ via the pushforward measure:

$$P_{\tilde{X}|x} = g_* P_{\tilde{Z}|z} \tag{1}$$

where $g_*$ is the pushforward operator. Our analysis focuses on how $P_{\tilde{Z}|z}$ affects the recovery map $h := f \circ g : \mathcal{Z} \to \mathcal{Z}$. The sampling mechanism need not be an image augmentation; it may also arise from temporal proximity, spatial crops, multimodal co-occurrence, or any other rule for draw-

ing related observations. Independent views mean conditionally independent draws from $P_{\tilde{Z}|z}$ for a fixed anchor $z$, not semantic independence of the resulting observations. Notation: uppercase $P, Q$ denotes probability laws, while lowercase $p, q$ denotes probability measure densities with respect to the stated reference measure.

### 3.3. Diversity Condition

We formalize the diversity condition and argue for its necessity in distance-preserving latent space reconstruction. The condition is motivated by comparing the theoretical ideal of full-support positive-pair sampling with practical sampling mechanisms. In the ideal vMF setting used by (Zimmermann et al., 2021), positive pairs can in principle cover the entire latent space around an anchor, which provides enough information to recover pairwise geometry. Practical mechanisms often restrict this support by keeping some latent coordinates fixed while changing others. The diversity condition identifies the weakest support requirement needed for recovery: every latent region that has nonzero marginal probability must also be reachable by the conditional positive-pair distribution.

**Definition 3.1** (Diversity Condition)**.** For a latent measurable space $\mathcal{Z}$ with marginal probability measure $P_Z$ and conditional probability measure $P_{\tilde{Z}|z}$, the diversity condition holds if, for $P_Z$-almost every $z \in \mathcal{Z}$,

$$P_Z \ll P_{\tilde{Z}|z}. \tag{2}$$

Equivalently, for every measurable set $A \subseteq \mathcal{Z}$, $P_{\tilde{Z}|z}(A) = 0$ implies $P_Z(A) = 0$ for $P_Z$-almost every anchor $z$.

Intuitively, the diversity condition requires that $P_{\tilde{Z}|z}$ has sufficiently large support to cover any region where $P_Z$ assigns nonzero probability. This implies that the view-generating process must perturb all latent features. If some generative features remain constant, the encoder cannot distinguish them along fixed dimensions, failing to invert $g$ for those components.

Finally, we relate the condition to practical sampling mechanisms. The diversity condition is stated at the level of the induced latent sampling mechanism $P_{\tilde{Z}|z}$, rather than in terms of application-specific categories of transformations. Descriptions such as appearance, structure, or semantic content are therefore only informal indicators of which latent directions a sampling mechanism may vary. In practice, an augmentation or view-generation rule is useful for recovery only to the extent that it gives the conditional distribution support along the latent factors present under $P_Z$. Mechanisms that vary only a restricted subset of these factors may violate the condition, even if they produce visually distinct observations.

### 3.4. InfoNCE as Cross-Entropy Minimization

Information Noise Contrastive Estimation (InfoNCE) (van den Oord et al., 2019) is the standard contrastive objective used to train representations from one positive view and a set of negative samples. For each anchor, it increases similarity to the positive sample while decreasing similarity to negatives, making it a probabilistic discrimination loss over co-occurring and non-co-occurring samples. The InfoNCE objective quantifies encoder performance on this discrimination task.

**Definition 3.2** (InfoNCE Loss (van den Oord et al., 2019))**.** Given a recovery map $h : \mathcal{Z} \to \mathcal{Z}$, positive pairs $(z, \tilde{z}) \sim P_{\text{pos}}$, and negative samples $\{z_i^-\}_{i=1}^{M} \overset{\text{i.i.d.}}{\sim} P^-$:

$$\mathcal{L}_{\text{CL}}(h; \tau, M) = \underset{\substack{(z,\tilde{z}) \sim P_{\text{pos}} \\ \{z_i^-\}_{i=1}^{M} \overset{\text{i.i.d.}}{\sim} P^-}}{\mathbb{E}} \left[ -\log \frac{e^{h(z)^{\top} h(\tilde{z})/\tau}}{D(z, \tilde{z})} \right]$$

$$D(z, \tilde{z}) = e^{h(z)^{\top} h(\tilde{z})/\tau} + \sum_{i=1}^{M} e^{h(z)^{\top} h(z_i^-)/\tau}$$

where $\tau > 0$ is a temperature parameter and $M$ is the number of negative samples.

Following (Zimmermann et al., 2021), we take the latent space to be a hypersphere $\mathbb{S}^{k-1}$, motivated by the practical convention of $L^2$-normalizing contrastive representations (Chen et al., 2020; Haas et al., 2024). We assume the conditional follows a von Mises-Fisher (vMF) distribution with concentration $\kappa > 0$, where sampling frequency is inversely proportional to latent distance.

**Theorem 3.3** (Asymptotics of $\mathcal{L}_{\text{CL}}$ (Zimmermann et al., 2021))**.** *Given a spherical latent space $\mathcal{Z} = \mathbb{S}^{k-1}$, a uniform marginal law $P_Z = U(\mathcal{Z})$, a von Mises-Fisher conditional measure $P_{\tilde{Z}|z}$ with density*

$$p(\tilde{z}|z) = \frac{e^{\kappa \tilde{z}^{\top} z}}{\int_{\mathcal{Z}} e^{\kappa z'^{\top} z} \, d\sigma(z')} \tag{3}$$

*where $\sigma$ denotes spherical surface measure on $\mathcal{Z}$ and $\kappa > 0$ is the concentration parameter,*

*and a model conditional measure $Q_{h,z}$ with density*

$$q_h(\tilde{z}|z) = \frac{e^{h(\tilde{z})^{\top} h(z)/\tau}}{\int_{\mathcal{Z}} e^{h(z')^{\top} h(z)/\tau} \, d\sigma(z')} \tag{4}$$

*For fixed $\tau > 0$, as the number of negative samples $M \to \infty$, the (normalized) contrastive loss converges to*

$$\lim_{M \to \infty} \mathcal{L}_{CL}(h; \tau, M) - \log M + \log |\mathcal{Z}| =$$
$$\mathbb{E}_{z \sim P_Z}[H(P_{\tilde{Z}|z}, Q_{h,z})] \tag{5}$$

where $H(P_{\tilde{Z}|z}, Q_{h,z})$ denotes cross-entropy, using densities with respect to $\sigma$ in the vMF setting.

This interpretation allows us to analyze contrastive learning through the lens of distribution matching between the sampling mechanism $P_{\tilde{Z}|z}$ and the model measure $Q_{h,z}$.

### 3.5. Inductive Bias

Inductive bias refers to the structural assumptions that constrain a learning algorithm's hypothesis space, enabling generalization (Vapnik, 1999). In contrastive learning, these assumptions arise through model architecture (e.g., translation equivariance in CNNs (LeCun et al., 1998), attention in Transformers (Dosovitskiy et al., 2021)) and geometric constraints on the embedding space. In our framework, an encoder is geometry-preserving when the recovery map $h = f \circ g$ recovers the latent structure up to an orthogonal transformation, i.e., $h(z) \approx Az$ for some $A \in O(k)$. Equivalently, the encoder approximates $f \approx A \circ g^{-1}$ on the data manifold. An effective inductive bias therefore makes approximate inverses of the data-generating process easier to represent and optimize, while restricting arbitrary non-geometry-preserving maps that can also satisfy the contrastive loss.

When the diversity condition is violated, the contrastive objective alone cannot uniquely determine the latent geometry. Inductive bias then acts as a compensatory mechanism, restricting admissible solutions to those consistent with architectural priors. In Section 4.3, we show that such biases are necessary for linearly identifiable reconstruction when diversity is violated. Crucially, we demonstrate experimentally (Section 5) that this necessity persists even asymptotically: the contrastive objective fails to recover latent geometry regardless of data quantity unless structural constraints are imposed.

## 4. Theoretical Results

We present the core theoretical results under the common assumption of $L^2$-normalized representations (Grill et al., 2020; Haas et al., 2023). Unless stated otherwise, the results use the following standing assumptions:

(A1) $\mathcal{Z} = \mathbb{S}^{k-1}$ is a unit hypersphere;
(A2) the marginal $P_Z$ is uniform on $\mathcal{Z}$;
(A3) the conditional $P_{\tilde{Z}|z}$ is vMF with concentration $\kappa > 0$ before diversity violation is introduced;
(A4) the encoder has sufficient capacity to realize any measurable recovery map considered below;
(A5) representations are $L^2$-normalized, so $h : \mathcal{Z} \to \mathcal{Z}$.

The full-support result below is closest to prior hypersphere identifiability results, but the proof uses a probabilistic

Mazur-Ulam theorem and requires no differentiability of $h$. The subsequent results are the main extension: they characterize the asymptotic global optima when sampling diversity is violated. We introduce the following definitions to formalize our analysis.

**Definition 4.1** (Isometry Almost Everywhere). Let $(\mathcal{Z}, \delta)$ be a metric space with measure $\mu$. A measurable mapping $h : \mathcal{Z} \to \mathcal{Z}$ is an *isometry almost everywhere* if there exists an isometry $e : \mathcal{Z} \to \mathcal{Z}$ such that $h(z) = e(z)$ for $\mu$-almost all $z \in \mathcal{Z}$.

**Definition 4.2** (Equivalent Recovery Maps). Two mappings $h_1, h_2 : \mathcal{Z} \to \mathcal{Z}$ are equivalent with respect to the contrastive loss if $\mathcal{L}_{\text{CL}}(h_1; \tau, M) = \mathcal{L}_{\text{CL}}(h_2; \tau, M)$ for fixed $\tau$ and $M$.

### 4.1. Reconstruction Under Full-Support Sampling

**Lemma 4.3** (Full-Support vMF Implies Diversity). *Let $P_Z$ be uniform on $\mathcal{Z} = \mathbb{S}^{k-1}$, and let $P_{\tilde{Z}|z}$ have vMF density proportional to $\exp(\kappa z^\top \tilde{z})$ with $\kappa > 0$ with respect to spherical surface measure. Then the diversity condition holds.*

*Proof.* The vMF density is strictly positive on the whole sphere. Therefore, any measurable set with zero $P_{\tilde{Z}|z}$ measure also has zero spherical surface measure, and hence zero uniform marginal measure. $\square$

Thus in the vMF setting the diversity condition is a consequence of full support, not an additional independent hypothesis. Contrastive learning then recovers the latent space up to orthogonal transformation.

**Theorem 4.4** (Linear Identifiability Under Full Diversity). *Under a uniform marginal on $\mathcal{Z} = \mathbb{S}^{k-1}$ and full-support vMF conditional, any recovery map $h : \mathcal{Z} \to \mathcal{Z}$ that globally minimizes the asymptotic contrastive objective is an isometry almost everywhere:*

$$h(z) = Az \quad \text{for } \mu\text{-almost all } z \in \mathcal{Z},$$

*where $A \in O(k)$ is an orthogonal matrix.*

*Proof.* See Appendix A.1. $\square$

*Proof sketch.* The asymptotic InfoNCE objective reduces to conditional cross-entropy. Under the full-support vMF conditional, any minimizer must match conditionals and therefore preserve inner products almost surely. On the sphere, this gives distance preservation, and the probabilistic Mazur-Ulam theorem extends this almost-sure isometry to a global orthogonal map.

This represents an ideal scenario: when the sampling mechanism perturbs all latent factors, the learned representations preserve pairwise distances, yielding a separable latent

space suitable for downstream tasks. However, real-world augmentation pipelines rarely have full support, motivating the analysis in the following section.

## 4.2. Reconstruction Under Violated Diversity

In practice, the diversity condition is rarely satisfied. Most augmentation pipelines preserve certain semantic features while perturbing others. We model this by decomposing the latent vector $z = (u, v)$ into an invariant component $u \in \mathbb{R}^m$ and a varying component $v \in \mathbb{R}^\ell$, with $m + \ell = k$ and $m, \ell > 0$. Let

$$K(z) := \{\tilde{z} = (\tilde{u}, \tilde{v}) \in \mathbb{S}^{k-1} : \tilde{u} = u\},$$
$$K(z) \cong \mathbb{S}^{\ell-1}_{r(z)}, \qquad r(z) := \sqrt{1 - \|u\|^2}.$$

For $r(z) > 0$, let $\sigma_{K(z)}$ denote the intrinsic surface measure on $K(z)$. The constrained positive-pair conditional is the probability measure $P^K_{\tilde{Z}|z}$ supported on $K(z)$ with Radon-Nikodym density

$$\frac{dP^K_{\tilde{Z}|z}}{d\sigma_{K(z)}}(\tilde{z}) = \frac{e^{\kappa z^\top \tilde{z}}}{\int_{K(z)} e^{\kappa z^\top z'} d\sigma_{K(z)}(z')}. \qquad (6)$$

This measure is singular with respect to ambient spherical measure on $\mathbb{S}^{k-1}$, but it is a well-defined probability law on the lower-dimensional submanifold $K(z)$.

This collapses the sampling support from $\mathbb{S}^{k-1}$ onto a lower-dimensional submanifold $K(z)$, mirroring the content-style framework of (von Kügelgen et al., 2021).

**Theorem 4.5** (Loss of Identifiability Under Violated Diversity). *For the asymptotic contrastive objective obtained as $M \to \infty$ under the constrained conditional (Equation 6) and standard full-sphere negatives, orthogonal recovery is not globally optimal. There exists a recovery map $h : \mathcal{Z} \to \mathcal{Z}$ that is not induced by any orthogonal transformation such that*

$$\mathcal{L}_{\mathrm{CL}}(h) < \mathcal{L}_{\mathrm{CL}}(\tilde{h}), \qquad \forall \tilde{h} \in O(k).$$

This is a statement about global values of the limiting objective, not about finite-sample optimization dynamics.

*Proof.* See Appendix A.2. □

*Proof sketch.* The constrained conditional fixes $u$ and only varies $v$, while the standard model conditional still normalizes over the full sphere. We construct an explicit non-orthogonal map $h_\lambda(u, v) = (u, \lambda v)/\|(u, \lambda v)\|$ with $\lambda < 1$ close to one. Because positive pairs share $u$, this map improves the alignment term to first order, while the uniformity term is stationary at the identity, yielding lower asymptotic loss than any orthogonal map.

The consequence is concrete: arbitrarily expressive encoders are actively disincentivized from learning orthogonal recovery maps when the diversity condition is violated. The contrastive objective rewards geometry-distorting solutions, leading to poorly structured latent spaces where semantic relationships are not preserved. Consequently, downstream tasks that rely on meaningful distance relationships in the representation space face a fundamental bottleneck that cannot be overcome by increasing encoder capacity alone.

## 4.3. Correcting the Model

The support mismatch between $P^K_{\tilde{Z}|z}$ and the standard full-sphere model conditional prevents the InfoNCE objective from favoring isometric solutions. We address this by constraining the model conditional to the same submanifold $K(z)$:

$$q^K_{h,z}(\tilde{z}) = \frac{e^{h(z)^\top h(\tilde{z})/\tau}}{\int_{K(z)} e^{h(z)^\top h(z')/\tau} d\sigma_{K(z)}(z')}, \qquad \tilde{z} \in K(z). \qquad (7)$$

Equivalently, for a positive pair $(z, \tilde{z})$ and negatives $z^-_1, \ldots, z^-_M \overset{\text{i.i.d.}}{\sim} U(K(z))$, the per-anchor adapted loss is

$$\ell_{\mathrm{adapt}} = -\log \frac{\exp(h(z)^\top h(\tilde{z})/\tau)}{D_a(z, \tilde{z})},$$
$$D_a(z, \tilde{z}) = \exp(h(z)^\top h(\tilde{z})/\tau)$$
$$+ \sum_{j=1}^M \exp(h(z)^\top h(z^-_j)/\tau). \qquad (8)$$

This corresponds to drawing negative samples from $K(z)$ rather than the full latent space. Exact sampling requires access to the invariant component $u$, which is generally unavailable in real data. In practice, same-anchor augmentations provide a proxy: independent transformations of the same anchor preserve the components fixed by the augmentation mechanism while varying the remaining components. Appendix C gives the resulting training step, but the adapted loss should be read primarily as a theoretical diagnostic rather than a fully practical replacement for standard InfoNCE.

**Theorem 4.6** (Orthogonal Mappings as Minimizers). *Under the corrected model (Equation 7), any orthogonal transformation $h \in O(k)$ minimizes the asymptotic contrastive loss.*

*Proof.* See Appendix A.3. □

*Proof sketch.* Once the model conditional is restricted to the same support $K(z)$ as the true conditional, any orthogonal map preserves the inner products that define the vMF density on that support. Thus the model conditional matches the

true conditional when $\kappa = 1/\tau$, minimizing cross-entropy. This proves achievability of orthogonal solutions, but not uniqueness.

Although the corrected objective admits orthogonal solutions, it does not guarantee that all minimizers are orthogonal. Thus, the adapted objective should not be interpreted as a fully general solution: it removes the support mismatch and makes geometry-preserving recovery achievable, but it does not make such recovery unique. The sampling mechanism determines which solutions are achievable, while inductive bias influences which solution is selected during optimization. This non-uniqueness highlights the necessity of inductive bias for selecting geometry-preserving solutions.

## 5. Experimental Results

### 5.1. Synthetic Dataset Experiments

We validate the theoretical predictions from Section 4 using a controlled synthetic setup with a spherical latent space $\mathcal{Z} = \mathbb{S}^2$.[1] Following the assumptions of our theoretical framework, the marginal law is uniform over the sphere, and positive pairs are sampled according to a von Mises-Fisher (vMF) conditional law $P_{\tilde{Z}|z}$ with density $p(\tilde{z}|z) = \text{vMF}(z, \kappa)$ and concentration parameter $\kappa = 1/\tau$ (see Equation 17), where $\tau$ is the temperature in the InfoNCE loss. The generative processes used in our experiments are illustrated in Figure 2. Details of the sampling procedures are provided in Appendix C.

**Experimental Setup.** We evaluate five generative processes $g : \mathcal{Z} \to \mathcal{X}$ of varying complexity: Identity, injective Linear map ($\mathbb{S}^2 \to \mathbb{R}^7$), Spiral rotation, Patches, and invertible MLP (Hyvärinen & Morioka, 2016). Detailed definitions are provided in Appendix B.1. For each generative process, we compare two encoder architectures: an MLP encoder with hidden dimensions $[128, 256, 256, 256, 128]$ representing low inductive bias, and an inverse encoder designed to invert the corresponding generative process, representing high inductive bias. The invertible MLP is an exception, as it lacks a strict analytic inverse; we therefore evaluate only the MLP encoder for this generative process. All experiments use InfoNCE loss with temperature $\tau = 0.3$, Adam optimizer (Kingma & Ba, 2015) (lr = $10^{-3}$), batch size 2000, and 5000 iterations. The generative process $g$ remains frozen throughout training, and each configuration is evaluated over 5 independent runs.

**Evaluation Metrics.** We assess reconstruction quality using linear identifiability ($R^2$), which measures recovery up

to affine transformation as in (Hyvärinen & Morioka, 2016). Additional metrics (Mean Correlation Coefficient and Angular Preservation Error) are provided in Appendix D.

**Results.** We summarize the experimental results in Table 1 and Figure 5.

**Diversity Condition Holds.** When the diversity condition is satisfied, the MLP encoder achieves near-perfect linear identifiability ($R^2 \geq 0.99$) across all five generative processes, confirming the theoretical prediction of Theorem 4.4. This demonstrates that a sufficiently expressive encoder can recover a representation linearly identifiable with the ground-truth latent space when the diversity condition holds, across varying degrees of injective generative processes.

**Diversity Condition Violated.** When the diversity condition is violated by fixing the first latent dimension during positive pair sampling, MLP performance collapses dramatically. Linear identifiability drops to $R^2 \in [0.05, 0.13]$ for Identity, MLP, and Linear generative processes, and $R^2 = 0.25$ for Patches. The Spiral process is an exception ($R^2 = 0.72$), maintaining poor but not catastrophic performance with high variance across runs. This validates Theorem 4.5: the contrastive objective alone no longer incentivizes geometry-preserving solutions when the conditional sampling support is restricted.

**Inductive Bias Compensation.** Under violated diversity, incorporating inductive bias through inverse encoders restores near-perfect recovery ($R^2 \geq 0.88$), with Identity, Linear, and Spiral processes achieving $R^2 \geq 0.99$. The inverse encoders, designed to mirror the structure of each generative process, successfully recover the latent geometry even when the sampling mechanism provides insufficient information. This demonstrates that architectural constraints can compensate for deficiencies in the sampling regime, highlighting the complementary roles of data augmentation and model design in contrastive learning.

**Adapted InfoNCE Loss.** The adapted InfoNCE loss partially recovers MLP performance under violated diversity ($R^2 \approx 0.60$–$0.65$), representing a substantial improvement over standard InfoNCE ($R^2 \approx 0.05$–$0.25$ for most processes). However, this falls short of the performance achieved with appropriate inductive bias ($R^2 \geq 0.88$). This confirms Theorem 4.6: correcting the model conditional makes isometric solutions achievable but does not guarantee them. From a practical standpoint, these results suggest that while loss modifications can mitigate the effects of violated diversity, incorporating architectural priors remains the more effective strategy, aligning with the empirical success of high inductive bias architectures such as CNNs and

---

[1]Code is available at https://github.com/BosonicJustin/CLTheory.

**(a) Identity**     **(b) Linear**     **(c) Spiral**     **(d) Patches**     **(e) Invertible MLP**

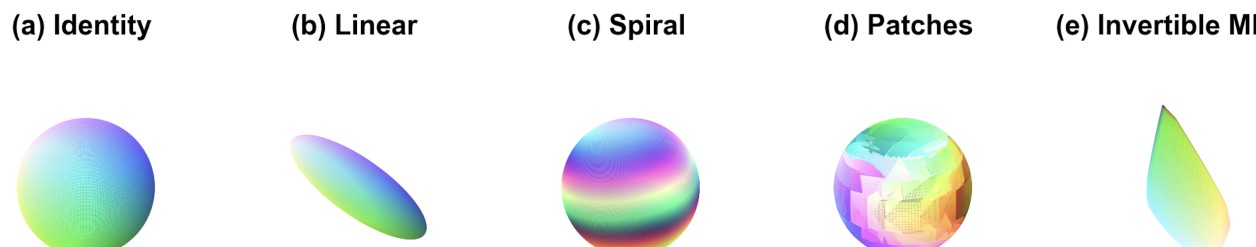

*Figure 2.* Generative processes mapping the unit sphere to observation space. Colors encode input coordinates (RGB = xyz), illustrating how each transformation warps the latent space: (a) identity preserves the sphere, (b) linear maps to an ellipsoid, (c) spiral twists points around the vertical axis, (d) patches applies piecewise rotations creating discontinuities, and (e) invertible MLP produces smooth nonlinear deformations.

Vision Transformers in contrastive learning pipelines.

### 5.2. CIFAR-10 Experiments

We test the qualitative implications of our theory on CIFAR-10 using SimCLR (Chen et al., 2020) with three encoder architectures of comparable size ($\sim$11M parameters each). We rate inductive bias by how closely each architecture's structural priors align with the spatial generative structure of natural images. Equivalently, this measures how strongly the architecture favors approximate inverses of image formation. ResNet-18 (He et al., 2016) has high inductive bias because convolutional layers encode spatial locality and translation equivariance. Vision Transformer (ViT) (Dosovitskiy et al., 2021) with $4 \times 4$ patches has medium inductive bias because self-attention can learn global spatial structure, but does not hardcode locality to the same extent. The MLP has low inductive bias because it treats each image as a flat vector and imposes little domain-specific restriction on the hypothesis space. All encoders project to 512-dimensional $L^2$-normalized embeddings and are initialized with random weights. Because CIFAR-10 does not provide ground-truth latent coordinates, linear probe accuracy is not a direct test of identifiability. We use it as a real-data sanity check for the qualitative predictions of the theory: richer sampling mechanisms and better aligned architectural inductive bias should improve downstream representation quality.

**Augmentation Regimes.** We design three augmentation regimes to vary the degree to which the diversity condition is approximated (Figure 4): (1) **All**: color jitter, random crop, horizontal flip, grayscale, blur, and cutout (DeVries & Taylor, 2017), perturbing as many features as possible; (2) **Crop Only**: Perturbing all features slightly, but with a lesser degree than All augmentations; (3) **All w/o crop**: all augmentations except crop, varying the features in a more fixed manner. Training uses InfoNCE with $\tau = 0.5$, Adam optimizer (Kingma & Ba, 2015) (lr = $3 \times 10^{-4}$), batch size 2000, for 200 epochs. We evaluate via linear probing over 5 runs per configuration.

**Results.** The results (Figure 3) are consistent with the qualitative predictions of the theory. First, the "All" augmentation regime yields the highest accuracy across all architectures, consistent with the expectation that broader sampling support improves representation quality. Second, higher inductive bias consistently improves performance: ResNet-18 outperforms ViT, which outperforms MLP, across all augmentation regimes. Third, and most importantly, the performance gap between architectures widens as the diversity condition is increasingly violated. Under the "All" regime, the gap between ResNet-18 and MLP is moderate; under "All w/o crop", this gap increases substantially. This interaction effect supports the compensatory role of inductive bias: when sampling diversity is insufficient, architectural priors become critical for recovering useful representations. Notably, for MLP the "All" and "Crop" regimes yield nearly identical performance, suggesting that without appropriate inductive bias, the encoder cannot exploit the additional augmentations, since cropping provides the majority of the meaningful signal.

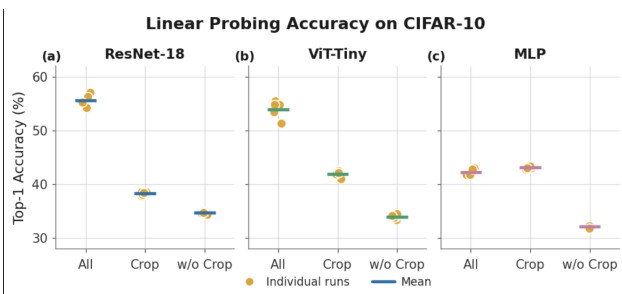

*Figure 3.* Linear probe accuracy on CIFAR-10 by architecture and augmentation regime. Individual runs shown as points; bars indicate mean $\pm 1$ std. The "All" regime best approximates the diversity condition and yields highest accuracy across all architectures.

### 6. Conclusion

We have presented a theoretical framework for understanding when contrastive learning recovers meaningful latent

*Table 1.* Linear identifiability ($R^2$) across generative processes under different experimental conditions. Results reported as mean $\pm$ std across 5 random seeds. The Invertible MLP process lacks a closed-form inverse, so no inverse encoder is available (indicated by N/A).

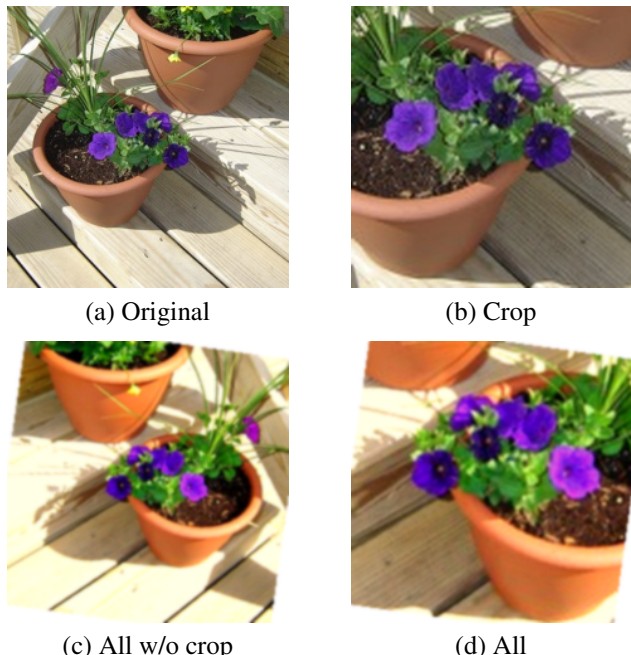

| | Diversity Holds | Diversity Violated | | |
|---|---|---|---|---|
| **Generative Process** | **InfoNCE** | **InfoNCE** | **InfoNCE Adapted** | **InfoNCE + Ind. Bias** |
| Identity | $1.00 \pm 0.00$ | $0.06 \pm 0.03$ | $0.63 \pm 0.00$ | $0.99 \pm 0.00$ |
| Invertible MLP | $1.00 \pm 0.00$ | $0.13 \pm 0.12$ | $0.60 \pm 0.01$ | N/A |
| Linear | $1.00 \pm 0.00$ | $0.05 \pm 0.06$ | $0.65 \pm 0.03$ | $0.99 \pm 0.00$ |
| Patches | $0.99 \pm 0.00$ | $0.25 \pm 0.03$ | $0.63 \pm 0.02$ | $0.88 \pm 0.01$ |
| Spiral | $1.00 \pm 0.00$ | $0.72 \pm 0.33$ | $0.62 \pm 0.01$ | $1.00 \pm 0.00$ |

(a) Original

(b) Crop

(c) All w/o crop

(d) All

*Figure 4.* CIFAR-10 augmentation regimes. (a) Original image. (b) Crop Only: random resized crop altering spatial extent. (c) All without crop: color jitter, horizontal flip, rotation, and blur. (d) All augmentations combined. Cropping changes visible spatial extent and local statistics, while color and blur transformations mainly alter appearance in this example.

representations. Our central contribution is the *diversity condition* (Definition 3.1), a requirement on $P_{\tilde{Z}|z}$ that is necessary for isometric latent recovery. When it holds, sufficiently expressive encoders recover the latent space up to an orthogonal transformation; when it is violated, the contrastive objective can actively disincentivize geometry-preserving solutions. The adapted InfoNCE objective makes isometric solutions achievable under violated diversity, but does not guarantee their selection. Our synthetic and CIFAR-10 experiments show that sampling diversity and architectural inductive bias jointly determine representation quality.

**Limitations and future work.** Motivated by practical $L^2$ normalization, our analysis uses a spherical latent space and vMF conditionals; future work should relax these assumptions and estimate diversity violation from data. Adapted InfoNCE is a theoretical diagnostic rather than a scalable objective: anchor-specific negatives require $O(N(M + 1))$ memory, and support correction restores achievability but not selection.

## Impact Statement

This work advances theoretical understanding of contrastive learning, a foundational technique for self-supervised representation learning. Our contributions are primarily theoretical, providing formal conditions (the diversity condition) under which contrastive methods succeed or fail at recovering meaningful latent structure.

The practical implications are indirect but potentially significant. By clarifying the interplay between sampling mechanisms and architectural inductive bias, our framework may guide more principled design of augmentation pipelines and encoder architectures, potentially reducing computational waste from trial-and-error experimentation. This could lower the environmental cost of training large-scale representation learning systems.

Contrastive learning underlies many deployed systems in vision, language, and multimodal domains. Improved theoretical understanding may help practitioners anticipate failure modes before deployment, particularly in high-stakes applications such as medical imaging or scientific discovery where representation quality directly affects downstream reliability.

We do not foresee direct negative societal consequences from this theoretical work. However, as with any advance in representation learning, improved methods could enhance both beneficial applications (e.g., drug discovery, climate modeling) and potentially harmful ones (e.g., surveillance). We encourage practitioners to consider the ethical implica-

tions of specific downstream applications enabled by better representation learning.

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

# A. Theoretical Results: Proofs

This appendix provides complete proofs for the theoretical results presented in Section 4. We organize the material into three subsections corresponding to the three scenarios analyzed: full diversity (Section A.1), violated diversity (Section A.2), and the corrected model (Section A.3).

## A.1. Proofs for Reconstruction Under Full Diversity

We first establish that cross-entropy minimizers preserve inner products almost surely, which forms the foundation for proving linear identifiability.

**Theorem A.1** (Cross-Entropy Minimizers Preserve Inner Products). *Let $\mathcal{Z} = \mathbb{S}^{k-1}$, $\kappa > 0$, and let $P_{\tilde{Z}|z}$ have density*

$$p(\tilde{z}|z) = C_p^{-1} \exp(\kappa \tilde{z}^\top z),$$

*where $C_p$ is the normalizing constant with respect to $\sigma$. Let $Q_{h,z}$ have density $q_h(\cdot|z)$ with respect to $\sigma$. Let $h : \mathcal{Z} \to \mathcal{Z}$ be a recovery map. If $h$ minimizes the cross-entropy*

$$\mathcal{L}_h := \mathbb{E}_{z \sim P_Z}[H(P_{\tilde{Z}|z}, Q_{h,z})],$$

*then $p(\tilde{z}|z) = q_h(\tilde{z}|z)$ $P_{\tilde{Z}|z}$-a.s., and $z^\top \tilde{z} = h(z)^\top h(\tilde{z})$ $P_{Z,\tilde{Z}}$-a.s.*

*Proof.* Since $h$ minimizes $\mathcal{L}_h$, it minimizes $H(P_{\tilde{Z}|z}, Q_{h,z})$ almost surely with respect to $P_Z$. The cross-entropy $H$ is minimized when

$$p(\cdot|z) = q_h(\cdot|z) \quad P_{\tilde{Z}|z}\text{-a.s.}$$

For a fixed $z$, this implies:

$$\frac{e^{\kappa \tilde{z}^\top z}}{C_p} = \frac{e^{h(\tilde{z})^\top h(z)/\tau}}{D(z)} \quad P_{\tilde{Z}|z}\text{-a.s.} \tag{9}$$

where $D(z)$ and $C_p$ are the normalizing constants for the respective von Mises-Fisher distributions.

Taking the logarithm yields:

$$\log\left(\frac{D(z)}{C_p}\right)\tau + \tau \kappa \tilde{z}^\top z = h(\tilde{z})^\top h(z) \quad P_{\tilde{Z}|z}\text{-a.s.} \tag{10}$$

Since $h$ maps onto the unit sphere,

$$-1 \le h(\tilde{z})^\top h(z) \le 1 \quad P_{\tilde{Z}|z}\text{-a.s.} \tag{11}$$

Rearranging gives:

$$-1 \le \log\left(\frac{D(z)}{C_p}\right)\tau + \tau \kappa \tilde{z}^\top z \le 1 \quad P_{\tilde{Z}|z}\text{-a.s.} \tag{12}$$

This simplifies to:

$$\frac{-1 - \log\left(\frac{D(z)}{C_p}\right)\tau}{\tau \kappa} \le \tilde{z}^\top z \le \frac{1 - \log\left(\frac{D(z)}{C_p}\right)\tau}{\tau \kappa} \tag{13}$$

Since $P_{\tilde{Z}|z}$ is a von Mises-Fisher distribution, the support of $\tilde{z}^\top z$ under this measure includes regions arbitrarily close to both 1 and $-1$. Specifically, for any $c \in (-1, 1)$:

$$\mu\left(\{\tilde{z} \in \mathcal{Z} : \tilde{z}^\top z \ge c\}\right) = \beta \int_0^{\cos^{-1}(c)} e^{\kappa \cos\theta} \sin^{k-2}\theta \, d\theta > 0 \tag{14}$$

where $\beta > 0$ is the spherical-measure normalization constant.

To satisfy the inequality almost surely with respect to $P_{\tilde{Z}|z}$, the bounds must equal the extremes:

$$\frac{1 - \log\left(\frac{D(z)}{C_p}\right)\tau}{\tau\kappa} = 1 \quad \text{and} \quad \frac{-1 - \log\left(\frac{D(z)}{C_p}\right)\tau}{\tau\kappa} = -1 \tag{15}$$

Both equations yield:

$$\log\left(\frac{D(z)}{C_p}\right) = 0 \implies D(z) = C_p \tag{16}$$

and

$$\kappa\tau = 1 \tag{17}$$

Substituting back:

$$\tilde{z}^\top z = h(\tilde{z})^\top h(z) \quad P_{\tilde{Z}|z}\text{-a.s.} \tag{18}$$

Since $h$ minimizes $\mathcal{L}_h$, this holds for almost all $z$ with respect to $P_Z$, implying:

$$\tilde{z}^\top z = h(\tilde{z})^\top h(z) \quad P_{Z,\tilde{Z}}\text{-a.s.} \tag{19}$$

$\square$

The following corollary translates inner product preservation to distance preservation, which is the key geometric property needed for identifiability.

**Corollary A.2** (Cross-Entropy Minimizers Preserve Distances Almost Everywhere). *If $h$ minimizes the expected cross-entropy loss $\mathbb{E}_{z \sim P_Z}[H(P_{\tilde{Z}|z}, Q_{h,z})]$, then*

$$\|h(z) - h(\tilde{z})\|^2 = \|z - \tilde{z}\|^2 \quad P_{Z,\tilde{Z}}\text{-a.s.}$$

*Proof.* Since both $z, \tilde{z} \in \mathcal{Z} = \mathbb{S}^{k-1}$ and $h(z), h(\tilde{z}) \in \mathcal{Z}' = \mathbb{S}^{k-1}$, we have $\|z\|^2 = \|\tilde{z}\|^2 = \|h(z)\|^2 = \|h(\tilde{z})\|^2 = 1$.

Expanding the squared Euclidean distance:

$$\|z - \tilde{z}\|^2 = \|z\|^2 - 2z^\top\tilde{z} + \|\tilde{z}\|^2 = 2 - 2z^\top\tilde{z} \tag{20}$$

Similarly, for the mapped points:

$$\|h(z) - h(\tilde{z})\|^2 = \|h(z)\|^2 - 2h(z)^\top h(\tilde{z}) + \|h(\tilde{z})\|^2 = 2 - 2h(z)^\top h(\tilde{z}) \tag{21}$$

By Theorem A.1, we have $z^\top\tilde{z} = h(z)^\top h(\tilde{z})$ $P_{Z,\tilde{Z}}$-a.s. Therefore:

$$\|h(z) - h(\tilde{z})\|^2 = 2 - 2h(z)^\top h(\tilde{z}) = 2 - 2z^\top\tilde{z} = \|z - \tilde{z}\|^2 \tag{22}$$

holds $P_{Z,\tilde{Z}}$-almost surely. $\square$

The distance preservation property established above holds only almost everywhere with respect to the joint distribution. To conclude that the optimal recovery map is a global orthogonal transformation, we apply the probabilistic generalization of the Mazur-Ulam theorem (Zaliaduonis & Gatidis, 2026), which shows that isometries holding almost everywhere on probability spaces can be extended to global isometries on the entire space.

*Proof of Theorem 4.4.* By Corollary A.2, the optimal recovery map $h : \mathcal{Z} \to \mathcal{Z}$ preserves distances almost everywhere with respect to the joint distribution $P_{Z,\tilde{Z}}$. Specifically, there exists a set $N \subset \mathcal{Z} \times \mathcal{Z}$ with $P_{Z,\tilde{Z}}(N) = 0$ such that

$$\|h(z) - h(\tilde{z})\| = \|z - \tilde{z}\| \quad \text{for all } (z, \tilde{z}) \in (\mathcal{Z} \times \mathcal{Z}) \setminus N.$$

Since the marginal law $P_Z$ is uniform on the sphere and has full support, $h$ preserves pairwise distances on a set of full $P_Z$-measure.

Applying the probabilistic Mazur-Ulam theorem (Zaliaduonis & Gatidis, 2026), which extends almost-everywhere isometries on probability spaces to global isometries, there exists $H : \mathbb{R}^k \to \mathbb{R}^k$ of the form $H(x) = Ax + b$ with $A \in O(k)$ and $b \in \mathbb{R}^k$, such that $h(z) = H(z)$ for $P_Z$-almost all $z \in \mathcal{Z}$.

Since both $\mathcal{Z}$ and $\mathcal{Z}'$ are unit spheres centered at the origin, and $H$ agrees almost everywhere with $h : \mathcal{Z} \to \mathcal{Z}'$, the translation component must vanish ($b = 0$) and the linear part must preserve the unit sphere. This implies that $A$ is an orthogonal matrix, i.e., $A^\top A = I$.

Therefore, the optimal recovery map $h$ coincides with an orthogonal transformation almost everywhere:

$$h(z) = Az \quad \text{for } \mu\text{-almost all } z \in \mathcal{Z},$$

where $A \in O(k)$ is an orthogonal matrix. $\qquad\square$

### A.2. Proofs for Reconstruction Under Violated Diversity

When the diversity condition is violated, the conditional law is constrained to a lower-dimensional submanifold. We decompose each latent vector $z \in \mathcal{Z} = \mathbb{S}^{k-1}$ into two components:

$$z = (u, v)^\top \in \mathbb{R}^m \times \mathbb{R}^\ell, \quad \text{where } m + \ell = k \text{ and } m, \ell > 0.$$

Here, $u \in \mathbb{R}^m$ represents latent dimensions that remain fixed under conditional sampling, while $v \in \mathbb{R}^\ell$ represents dimensions that can vary. Let $K(z) = \{\tilde{z} \in \mathbb{S}^{k-1} : \tilde{u} = u\} \cong \mathbb{S}^{\ell-1}_{r(z)}$, where $r(z) = \sqrt{1 - \|u\|^2}$. Let $\sigma_{K(z)}$ denote intrinsic surface measure on $K(z)$. The constrained conditional is the probability measure $P^K_{\tilde{Z}|z}$ with density

$$\frac{dP^K_{\tilde{Z}|z}}{d\sigma_{K(z)}}(\tilde{z}) = \frac{e^{\kappa \tilde{z}^\top z}}{\int_{K(z)} e^{\kappa z^\top z'} d\sigma_{K(z)}(z')}. \tag{23}$$

It is singular with respect to ambient spherical measure on $\mathbb{S}^{k-1}$, but absolutely continuous with respect to the intrinsic measure on $K(z)$.

Despite this modification, the asymptotic relationship between the contrastive loss and cross-entropy minimization remains valid.

**Theorem A.3** (Asymptotic Equivalence Under Violated Diversity). *Given the constrained conditional probability measure defined above, the marginal law $P_Z$ uniform on $\mathcal{Z} = \mathbb{S}^{k-1}$, temperature $\tau > 0$, and number of negative samples $M > 0$, as $M \to \infty$:*

$$\lim_{M \to \infty} \mathcal{L}_{CL}(h; \tau, M) - \log M + \log |\mathcal{Z}| = \mathbb{E}_{z \sim P_Z}[H(P^K_{\tilde{Z}|z}, Q_{h,z})]$$

*Proof.* The proof follows identically to Theorem 3.3, as the asymptotic analysis relies only on the law of large numbers and properties of the logarithm, not on the specific structure of the ground-truth conditional law. See (Wang & Isola, 2020) for the full argument. $\qquad\square$

The critical distinction arises when comparing global values of the limiting objective. Under violated diversity, orthogonal recovery maps are not global minimizers.

*Proof of Theorem 4.5.* Let $\sigma$ denote the normalized spherical measure on $\mathcal{Z} = \mathbb{S}^{k-1}$. By Theorem A.3, it suffices to compare the asymptotic contrastive loss up to constants independent of $h$:

$$\mathcal{L}(h) = -\frac{1}{\tau}\mathbb{E}_{(z,\tilde{z}) \sim P_{\text{pos}}}\left[h(z)^\top h(\tilde{z})\right] + \mathbb{E}_{z \sim \sigma} \log \int_{\mathbb{S}^{k-1}} \exp\left(\frac{h(z)^\top h(z')}{\tau}\right) d\sigma(z').$$

For $\lambda > 0$, define

$$h_\lambda(u, v) := \frac{(u, \lambda v)}{\sqrt{\|u\|^2 + \lambda^2 \|v\|^2}}.$$

At $\lambda = 1$, $h_\lambda$ is the identity. For $\lambda \neq 1$, $h_\lambda$ is not orthogonal because it changes inner products between points with different relative invariant and variant components.

Write the alignment and uniformity terms as

$$A(\lambda) := \mathbb{E}_{(z,\tilde{z}) \sim P_{\mathrm{pos}}} \left[ h_\lambda(z)^\top h_\lambda(\tilde{z}) \right],$$

$$U(\lambda) := \mathbb{E}_{z \sim \sigma} \log \int_{\mathbb{S}^{k-1}} \exp\left( \frac{h_\lambda(z)^\top h_\lambda(z')}{\tau} \right) d\sigma(z').$$

Then $\mathcal{L}(h_\lambda) = -\tau^{-1} A(\lambda) + U(\lambda)$.

We first show that shrinking the variant component improves alignment to first order. Under the constrained positive-pair distribution, $\tilde{u} = u$. Since $z, \tilde{z} \in \mathbb{S}^{k-1}$, this implies $\|v\| = \|\tilde{v}\|$. Therefore

$$h_\lambda(z)^\top h_\lambda(\tilde{z}) = \frac{\|u\|^2 + \lambda^2 v^\top \tilde{v}}{\|u\|^2 + \lambda^2 \|v\|^2}.$$

With $t = \lambda^2$ and

$$f(t) = \frac{\|u\|^2 + t\, v^\top \tilde{v}}{\|u\|^2 + t\|v\|^2},$$

we have

$$f'(t) = \frac{\|u\|^2 (v^\top \tilde{v} - \|v\|^2)}{(\|u\|^2 + t\|v\|^2)^2}.$$

By Cauchy-Schwarz, $v^\top \tilde{v} \leq \|v\|^2$, with strict inequality whenever $v \neq \tilde{v}$. The constrained vMF conditional is non-degenerate for $\kappa > 0$, so $v \neq \tilde{v}$ on a set of positive $P_{\mathrm{pos}}$-measure, and $\|u\| > 0$ for $\sigma$-almost every $z$ when $m > 0$. Hence $A'(1) < 0$, and for some $c > 0$,

$$A(\lambda) - A(1) = c(1 - \lambda) + o(1 - \lambda) \qquad \text{as } \lambda \uparrow 1.$$

We next show that the uniformity term has zero first derivative at the identity. Let

$$g(z) := \left. \frac{d}{d\lambda} h_\lambda(z) \right|_{\lambda=1} = (-\|v\|^2 u, \, \|u\|^2 v),$$

so $z^\top g(z) = 0$. At $\lambda = 1$, the inner integral

$$C := \int_{\mathbb{S}^{k-1}} \exp(z^\top z'/\tau) d\sigma(z')$$

is independent of $z$ by rotational symmetry. Differentiating under the integral, which is justified by smoothness and compactness of the sphere,

$$U'(1) = \frac{1}{\tau C} \iint \exp(z^\top z'/\tau) \left( g(z)^\top z' + z^\top g(z') \right) d\sigma(z') d\sigma(z).$$

For any fixed $a \in \mathbb{S}^{k-1}$, rotational symmetry gives

$$\int_{\mathbb{S}^{k-1}} z \exp(a^\top z/\tau) d\sigma(z) = \alpha a$$

for some scalar $\alpha$. Applying this identity to each of the two terms above and using $z^\top g(z) = 0$ yields $U'(1) = 0$. Thus

$$U(\lambda) - U(1) = o(1 - \lambda) \qquad \text{as } \lambda \uparrow 1.$$

Combining the two expansions,

$$\mathcal{L}(h_\lambda) - \mathcal{L}(I) = -\frac{1}{\tau} \left( A(\lambda) - A(1) \right) + \left( U(\lambda) - U(1) \right) = -\frac{c}{\tau}(1 - \lambda) + o(1 - \lambda) < 0$$

for all $\lambda < 1$ sufficiently close to 1.

Finally, every orthogonal map $\tilde{h} \in O(k)$ preserves inner products and the uniform spherical measure, so $\mathcal{L}(\tilde{h}) = \mathcal{L}(I)$. Taking $h = h_\lambda$ for any $\lambda < 1$ sufficiently close to 1 gives

$$\mathcal{L}_{\mathrm{CL}}(h) < \mathcal{L}_{\mathrm{CL}}(\tilde{h}), \qquad \forall \tilde{h} \in O(k),$$

in the asymptotic regime $M \to \infty$. $\qquad\square$

### A.3. Proofs for the Corrected Model

To address the support mismatch, we modify the model conditional to incorporate the same constraint structure:

$$q_{h,z}^K(\tilde{z}) = \frac{e^{h(z)^\top h(\tilde{z})/\tau}}{\int_{K(z)} e^{h(z)^\top h(z')/\tau} d\sigma_{K(z)}(z')}, \qquad \tilde{z} \in K(z). \tag{24}$$

This modification ensures that $\mathrm{supp}(Q_{h,z}^K) = \mathrm{supp}(P_{\tilde{Z}|z}^K)$, eliminating the support mismatch.

**Theorem A.4** (Asymptotic Form of Modified Contrastive Loss). *Under the corrected model conditional, where negative samples are drawn uniformly from the constrained manifold $K(z)$ rather than from the full sphere $\mathcal{Z}$, the asymptotic contrastive loss takes the form:*

$$\lim_{M \to \infty} \mathcal{L}(h, \tau, M) - \log(M) = \mathbb{E}_{z \sim P_Z}[H(P_{\tilde{Z}|z}^K, Q_{h,z}^K)] - \mathbb{E}_{z \sim P_Z}[\log(|K(z)|)]$$

The proof follows the same steps as in (Zimmermann et al., 2021), but with conditionals defined on $K(z)$.

*Proof.* **Step 1: Cross-entropy decomposition.** The cross-entropy between the true and model conditionals, both defined on $K(z)$, is:

$$H(P_{\tilde{Z}|z}^K, Q_{h,z}^K) = -\mathbb{E}_{\tilde{z} \sim P_{\tilde{Z}|z}^K}[\log q_{h,z}^K(\tilde{z})] \tag{25}$$

$$= -\mathbb{E}_{\tilde{z} \sim P_{\tilde{Z}|z}^K}\left[\log\left(\frac{e^{h(z)^\top h(\tilde{z})/\tau}}{\int_{K(z)} e^{h(z)^\top h(z')/\tau} d\sigma_{K(z)}(z')}\right)\right]. \tag{26}$$

$$H(P_{\tilde{Z}|z}^K, Q_{h,z}^K) = -\mathbb{E}_{\tilde{z} \sim P_{\tilde{Z}|z}^K}\left[\frac{1}{\tau}h(z)^\top h(\tilde{z}) - \log C_h(z)\right] \tag{27}$$

$$= -\frac{1}{\tau}\mathbb{E}_{\tilde{z} \sim P_{\tilde{Z}|z}^K}[h(z)^\top h(\tilde{z})] + \log C_h(z) \tag{28}$$

where $C_h(z) = \int_{K(z)} e^{h(z)^\top h(z')/\tau} d\sigma_{K(z)}(z')$ is the normalizing constant over the constrained manifold.

**Step 2: Normalizing constant estimation.** Using the fact that the uniform distribution on $K(z)$ has density $1/|K(z)|$:

$$C_h(z) = \int_{K(z)} e^{h(z)^\top h(z')/\tau} d\sigma_{K(z)}(z') = |K(z)| \cdot \mathbb{E}_{z' \sim \mathrm{U}(K(z))}\left[e^{h(z)^\top h(z')/\tau}\right] \tag{29}$$

**Step 3: Final form.** Substituting the estimate of $C_h(z)$ back into the cross-entropy expression and splitting the logarithm:

$$H(P_{\tilde{Z}|z}^K, Q_{h,z}^K) = -\frac{1}{\tau}\mathbb{E}_{\tilde{z} \sim P_{\tilde{Z}|z}^K}[h(z)^\top h(\tilde{z})] \tag{30}$$

$$+ \log \mathbb{E}_{z' \sim \mathrm{U}(K(z))}\left[e^{h(z)^\top h(z')/\tau}\right] + \log |K(z)| \tag{31}$$

Taking expectations over $z \sim P_Z$ yields the stated result. $\qquad\square$

*Proof of Theorem 4.6.* With the corrected model conditional, the supports of $P^K_{\tilde{Z}|z}$ and $Q^K_{h,z}$ now match: both are restricted to the submanifold $K(z)$. Within this constrained setting, the analysis proceeds analogously to Theorem A.1.

For any $h \in O(k)$, we have $h(z)^\top h(\tilde{z}) = z^\top \tilde{z}$ for all $z, \tilde{z} \in \mathcal{Z}$. This means the model conditional $Q^K_{h,z}$ exactly matches the true conditional $P^K_{\tilde{Z}|z}$ on the constrained manifold $K(z)$:

$$q^K_{h,z}(\tilde{z}) = \frac{e^{z^\top \tilde{z}/\tau}}{\int_{K(z)} e^{z^\top z'/\tau} d\sigma_{K(z)}(z')} = \frac{dP^K_{\tilde{Z}|z}}{d\sigma_{K(z)}}(\tilde{z})$$

when $\kappa = 1/\tau$.

Since matching distributions achieves zero KL divergence and hence minimal cross-entropy, any orthogonal transformation minimizes the asymptotic contrastive loss. $\qquad\square$

Although the corrected objective admits orthogonal solutions, it does not guarantee uniqueness. The following theorem shows that multiple equivalent solutions exist.

**Theorem A.5** (Equivalence of Feature Extractors Under Constrained Sampling). *Given a data-generating process $g : \mathcal{Z} \to \mathcal{X}$, uniform marginal law $P_Z$, ground-truth conditional law $P_{\tilde{Z}|z}$ with density $p(\tilde{z}|z)$, and model conditional law $Q_{h,z}$ with density $q_h(\tilde{z}|z)$ that define the expected cross-entropy loss*

$$\mathcal{L}_h = \mathbb{E}_{z \sim P_Z}[H(P_{\tilde{Z}|z}, Q_{h,z})],$$

*let $m : \mathcal{Z} \to \mathcal{Z}$ be an invertible mapping that preserves the marginal law and the ground-truth conditional density:*

$$m_\# P_Z = P_Z, \qquad p(m(\tilde{z})|m(z)) = p(\tilde{z}|z) \quad \forall z, \tilde{z} \in \mathcal{Z}.$$

*Then any two mappings $h_1 := f_1 \circ g$ and $h_2 := f_2 \circ g$ with $h_2(z) := h_1(m(z))$ are equivalent, i.e., $\mathcal{L}_{h_1} = \mathcal{L}_{h_2}$.*

*Proof.* Starting with the cross-entropy loss for $h_2$:

$$\mathcal{L}_{h_2} = \mathbb{E}_{z \sim P_Z}\left[\mathbb{E}_{\tilde{z} \sim P_{\tilde{Z}|z}}[-\log q_{h_2}(\tilde{z}|z)]\right] \tag{32}$$

Using the definition of $h_2$:

$$= \mathbb{E}_{z \sim P_Z}\left[\mathbb{E}_{\tilde{z} \sim P_{\tilde{Z}|z}}[-\log q_{h_1}(m(\tilde{z})|m(z))]\right] \tag{33}$$

Since $m$ preserves the conditional density:

$$= \mathbb{E}_{z \sim P_Z}\left[\mathbb{E}_{m(\tilde{z}) \sim P_{\tilde{Z}|m(z)}}[-\log q_{h_1}(m(\tilde{z})|m(z))]\right] \tag{34}$$

Since $m_\# P_Z = P_Z$, we can change variables in the outer expectation:

$$= \mathbb{E}_{m(z) \sim P_Z}\left[\mathbb{E}_{m(\tilde{z}) \sim P_{\tilde{Z}|m(z)}}[-\log q_{h_1}(m(\tilde{z})|m(z))]\right] \tag{35}$$

Finally, because $m$ is invertible:

$$= \mathbb{E}_{z \sim P_Z}\left[\mathbb{E}_{\tilde{z} \sim P_{\tilde{Z}|z}}[-\log q_{h_1}(\tilde{z}|z)]\right] = \mathcal{L}_{h_1} \tag{36}$$

$$\square$$

This theorem reveals a fundamental non-uniqueness in the solution space of the modified contrastive learning objective. Any conditional-preserving transformation of the latent space yields identical loss values, meaning that without additional constraints through inductive bias, the objective cannot distinguish between semantically meaningful representations and arbitrary rearrangements that preserve only local structure within constrained manifolds.

# B. Generative Processes and Encoder Architectures

This section describes the generative processes and encoder architectures used in our synthetic experiments.

## B.1. Generative Processes

We employ five generative processes $g : \mathbb{S}^{d-1} \to \mathbb{R}^D$ of varying complexity to test our theoretical predictions across different data-generating mechanisms.

---

**Algorithm 1** Identity Process

---

**Require:** $z \in \mathbb{S}^{d-1}$
1: $x \leftarrow z$
2: **return** $x \in \mathbb{R}^d$

---

**Algorithm 2** Linear Process

---

**Require:** $z \in \mathbb{S}^{d-1}$, weight matrix $W \in \mathbb{R}^{D \times d}$ with $\text{rank}(W) = d$
1: $x \leftarrow Wz$
2: **return** $x \in \mathbb{R}^D$

---

**Algorithm 3** Spiral Rotation Process

---

**Require:** $z = (z_1, z_2, z_3) \in \mathbb{S}^2$, period $n$
1: Compute rotation angle: $\theta \leftarrow n\pi z_3$
2: Apply 2D rotation to first two coordinates:
3:    $x_1 \leftarrow \cos(\theta)z_1 - \sin(\theta)z_2$
4:    $x_2 \leftarrow \sin(\theta)z_1 + \cos(\theta)z_2$
5: $x_3 \leftarrow z_3$
6: **return** $x = (x_1, x_2, x_3) \in \mathbb{R}^3$

---

**Algorithm 4** Patches Process

---

**Require:** $z = (z_1, z_2, z_3) \in \mathbb{S}^2$, number of slices $K$
1: **Step 1:** Apply piecewise rotation based on $z_3$
2:    Determine bucket $k \leftarrow \lfloor (z_3 + 1) \cdot K/2 \rfloor$
3:    Compute angle $\theta_k \leftarrow -\pi / \max(1, K - k)$
4:    $z' \leftarrow R_{xy}(\theta_k) \cdot z$ {Rotate in $(x, y)$ plane}
5: **Step 2:** Apply 3D rotation (pitch $= \pi/2$)
6:    $z'' \leftarrow R_y(\pi/2) \cdot z'$
7: **Step 3:** Apply second piecewise rotation
8:    Determine new bucket, apply rotation as in Step 1
9: **return** $x \in \mathbb{R}^3$

---

**Algorithm 5** Invertible MLP Process (Hyvärinen & Morioka, 2016)

---

**Require:** $z \in \mathbb{S}^{d-1}$, MLP layers $\{W_i, b_i\}_{i=1}^L$ with conditioning
1: $h_0 \leftarrow z$
2: **for** $i = 1$ to $L$ **do**
3:    $h_i \leftarrow \sigma(W_i h_{i-1} + b_i)$ {$\sigma = \text{LeakyReLU}$}
4: **end for**
5: **return** $x = h_L \in \mathbb{R}^D$

---

## B.2. Encoder Architectures

We compare two classes of encoders: a generic MLP encoder representing low inductive bias, and inverse encoders designed to mirror the structure of each generative process, representing high inductive bias.

---

**Algorithm 6** MLP Encoder (Low Inductive Bias)

---

**Require:** $x \in \mathbb{R}^D$, hidden dims $[128, 256, 256, 256, 128]$
1: $h_0 \leftarrow x$
2: **for** $i = 1$ to $L - 1$ **do**
3:    $h_i \leftarrow \text{ReLU}(\text{BatchNorm}(W_i h_{i-1} + b_i))$
4: **end for**
5: $z' \leftarrow W_L h_{L-1} + b_L$
6: $z \leftarrow z'/\|z'\|_2$ {Project to sphere}
7: **return** $z \in \mathbb{S}^{d-1}$

---

**Algorithm 7** Inverse Linear Encoder (High Inductive Bias)

---

**Require:** $x \in \mathbb{R}^D$, learnable $W \in \mathbb{R}^{d \times D}$, $b \in \mathbb{R}^d$
1: $z' \leftarrow Wx + b$
2: $z \leftarrow z'/\|z'\|_2$
3: **return** $z \in \mathbb{S}^{d-1}$

---

**Algorithm 8** Inverse Spiral Encoder (High Inductive Bias)

---

**Require:** $x \in \mathbb{R}^3$, period $n$
1: Predict rotation control: $c \leftarrow \text{MLP}_{\text{rot}}(x)$ {3-layer MLP}
2: Extract spatial components: $(x_1, x_2) \leftarrow x_{1:2}$
3: Compute inverse rotation: $\theta \leftarrow -n\pi c$
4: Apply inverse rotation:
5:    $z_1 \leftarrow \cos(\theta)x_1 - \sin(\theta)x_2$
6:    $z_2 \leftarrow \sin(\theta)x_1 + \cos(\theta)x_2$
7: $z' \leftarrow (z_1, z_2, c)$
8: $z \leftarrow z'/\|z'\|_2$
9: **return** $z \in \mathbb{S}^2$

---

**Algorithm 9** Inverse Patches Encoder (High Inductive Bias)

---

**Require:** $x \in \mathbb{R}^3$, number of slices $K$
1: Predict original $z$-coordinate: $z_{\text{pred}} \leftarrow \tanh(\text{MLP}_z(x))$
2: Predict bucket probabilities: $w \leftarrow \text{softmax}(\text{MLP}_{\text{bucket}}(x))$
3: **Step 1:** Inverse second piecewise rotation
4:    $x_1 \leftarrow \sum_{k=0}^{K-1} w_k \cdot R_{xy}(-\theta_k) \cdot x$ {Soft inverse}
5: **Step 2:** Inverse 3D rotation
6:    $x_2 \leftarrow R_y(-\pi/2) \cdot x_1$
7: **Step 3:** Inverse first piecewise rotation
8:    $x_3 \leftarrow \sum_{k=0}^{K-1} w_k \cdot R_{xy}(-\theta_k) \cdot x_2$
9: Replace $z$-coordinate: $x_{\text{rec}} \leftarrow (x_{3,1}, x_{3,2}, z_{\text{pred}})$
10: $z \leftarrow x_{\text{rec}}/\|x_{\text{rec}}\|_2$
11: **return** $z \in \mathbb{S}^2$

---

## C. Sampling Procedures

We use standard techniques for sampling from the uniform distribution on the sphere and the von Mises-Fisher distribution.

---

**Algorithm 10** Uniform Sampling from $\mathbb{S}^{d-1}$

---

**Require:** Dimension $d \in \mathbb{N}$
1: Draw $z_i \sim \mathcal{N}(0, 1)$ independently for $i = 1, \ldots, d$
2: $z \leftarrow (z_1, \ldots, z_d)^\top$
3: $v \leftarrow z / \|z\|_2$
4: **return** $v \in \mathbb{S}^{d-1}$

---

**Algorithm 11** von Mises-Fisher Sampling (Wood, 1994)

---

**Require:** Mean direction $\mu \in \mathbb{S}^{d-1}$, concentration $\kappa > 0$
1: $p \leftarrow d - 1$
2: $b \leftarrow p / (\sqrt{4\kappa^2 + p^2} + 2\kappa)$
3: $x \leftarrow (1 - b)/(1 + b)$
4: $c \leftarrow \kappa x + p \log(1 - x^2)$
5: **repeat**
6:     Sample $t \sim \text{Beta}(p/2, p/2)$
7:     $w \leftarrow (1 - (1 + b)t)/(1 - (1 - b)t)$
8:     Sample $u \sim \text{Uniform}(0, 1)$
9: **until** $\log(u) \leq \kappa w + p \log(1 - xw) - c$
10: Sample $g \sim \mathcal{N}(0, I_d)$
11: $v \leftarrow g - (g^\top \mu)\mu$ {Project out $\mu$ component}
12: $v \leftarrow v / \|v\|_2$
13: $s \leftarrow \sqrt{1 - w^2} \cdot v + w \cdot \mu$
14: **return** $s \sim \text{vMF}(\mu, \kappa)$

---

**Algorithm 12** Conditional Sampling: Diversity Holds

---

**Require:** Anchor $z \in \mathbb{S}^{d-1}$, concentration $\kappa$
1: $\tilde{z} \leftarrow \text{vMF}(z, \kappa)$ {Sample positive using Algorithm 11}
2: **return** $\tilde{z} \in \mathbb{S}^{d-1}$

---

**Algorithm 13** Conditional Sampling: Diversity Violated

---

**Require:** Anchor $z = (u, v) \in \mathbb{S}^{d-1}$, fixed dimensions $d_{\text{fixed}}$, concentration $\kappa$
1: $u \leftarrow z_{1:d_{\text{fixed}}}$ {Fixed component}
2: $v \leftarrow z_{d_{\text{fixed}}+1:d}$ {Varying component}
3: $r \leftarrow \|v\|_2$ {Radius of sub-sphere}
4: **if** $r > 0$ **then**
5:     $\hat{v} \leftarrow v/r$ {Normalize to sub-sphere}
6:     $\tilde{v} \leftarrow \text{vMF}(\hat{v}, \kappa)$ {Sample on sub-sphere}
7:     $\tilde{v} \leftarrow r \cdot \tilde{v}$ {Scale back}
8: **else**
9:     $\tilde{v} \leftarrow v$
10: **end if**
11: $\tilde{z} \leftarrow (u, \tilde{v})$ {Concatenate fixed and sampled}
12: **return** $\tilde{z} \in \mathbb{S}^{d-1}$

---

---

**Algorithm 14** Adapted InfoNCE with Same-anchor Negatives

---

**Require:** Encoder $h$, batch $\{x_1, \ldots, x_N\}$, stochastic augmentation $\mathcal{T}$, temperature $\tau$, negatives per anchor $M$
  1: **for** each anchor $x_i$ **do**
  2:   Draw positive view $\tilde{x}_i \leftarrow \mathcal{T}(x_i)$
  3:   **for** $j = 1$ to $M$ **do**
  4:     Draw same-anchor negative $x_{i,j}^- \leftarrow \mathcal{T}(x_i)$ independently
  5:   **end for**
  6:   Compute

$$\mathcal{L}_i = -\log \frac{\exp(h(x_i)^\top h(\tilde{x}_i)/\tau)}{\exp(h(x_i)^\top h(\tilde{x}_i)/\tau) + \sum_{j=1}^{M} \exp(h(x_i)^\top h(x_{i,j}^-)/\tau)}.$$

  7: **end for**
  8: **return** $\mathcal{L} = \frac{1}{N}\sum_{i=1}^{N}\mathcal{L}_i$

---

Algorithm 14 approximates sampling negatives from $K(z)$ in observation space. Each call to $\mathcal{T}$ uses independent randomness conditional on the same anchor $x_i$, which is the observation-space analogue of drawing conditionally independent samples from $P_{\tilde{Z}|z_i}$. The same sampling family is applied repeatedly to the same anchor, so all views preserve the invariant component encoded by the mechanism while varying the remaining components. This changes the negative-sampling task relative to standard InfoNCE, which samples negatives from other instances. In our framework this is consistent with the goal of recovering latent distance structure rather than discriminating instance identity, but it can trade cross-instance discrimination signal for better support matching. The approximation also samples from the augmentation-induced distribution on $K(z)$ rather than uniformly from $K(z)$.

## D. Evaluation Metrics

We evaluate latent space reconstruction quality using three complementary metrics, following standard practice in the identifiability literature (Hyvärinen & Morioka, 2016; Zimmermann et al., 2021).

---

**Algorithm 15** Linear Identifiability ($R^2$)

---

**Require:** Ground-truth latents $Z = \{z_i\}_{i=1}^n$, recovered latents $\hat{Z} = \{\hat{z}_i\}_{i=1}^n$
  1: Fit linear regression: $\hat{z} = Az + b$ minimizing $\sum_i \|z_i - (A\hat{z}_i + b)\|^2$
  2: Compute predictions: $\tilde{z}_i \leftarrow A\hat{z}_i + b$
  3: Compute total variance: $\text{SS}_{\text{tot}} \leftarrow \sum_{i=1}^n \|z_i - \bar{z}\|^2$
  4: Compute residual variance: $\text{SS}_{\text{res}} \leftarrow \sum_{i=1}^n \|z_i - \tilde{z}_i\|^2$
  5: $R^2 \leftarrow 1 - \text{SS}_{\text{res}}/\text{SS}_{\text{tot}}$
  6: **return** $R^2 \in (-\infty, 1]$ {1.0 = perfect linear recovery}

---

**Algorithm 16** Mean Correlation Coefficient (MCC)

---

**Require:** Ground-truth latents $Z \in \mathbb{R}^{n \times d}$, recovered latents $\hat{Z} \in \mathbb{R}^{n \times d}$
  1: Compute correlation matrix $C \in \mathbb{R}^{d \times d}$:
  2:   $C_{ij} \leftarrow |\text{corr}(Z_{:,i}, \hat{Z}_{:,j})|$
  3: Find optimal assignment via Munkres algorithm:
  4:   $\pi^* \leftarrow \arg\max_\pi \sum_{i=1}^d C_{i,\pi(i)}$
  5: $\text{MCC} \leftarrow \frac{1}{d}\sum_{i=1}^d C_{i,\pi^*(i)}$
  6: **return** $\text{MCC} \in [0, 1]$ {1.0 = perfect factor alignment}

---

---

**Algorithm 17** Angular Preservation Error (APE)

---

**Require:** Ground-truth latents $Z = \{z_i\}_{i=1}^n$, recovered latents $\hat{Z} = \{\hat{z}_i\}_{i=1}^n$
1: Initialize APE $\leftarrow 0$
2: **for** $i = 1$ to $n$ **do**
3:   **for** $j = 1$ to $n, j \neq i$ **do**
4:     APE $\leftarrow$ APE $+ |z_i^\top z_j - \hat{z}_i^\top \hat{z}_j|$
5:   **end for**
6: **end for**
7: APE $\leftarrow$ APE$/(n(n-1))$
8: **return** APE $\in [0, 2]$ {0.0 = perfect isometry}

---

| Metric | Range | Measures | Optimal |
|--------|-------|----------|---------|
| $R^2$ | $(-\infty, 1]$ | Linear predictability | 1.0 |
| MCC | $[0, 1]$ | Factor alignment (up to permutation) | 1.0 |
| APE | $[0, 2]$ | Angular/geometric preservation | 0.0 |

*Table 2.* Summary of evaluation metrics for latent space reconstruction.

## E. Additional Experimental Results

### E.1. Diversity Condition Holds

Table 3 reports MCC, APE, and final loss when the diversity condition is satisfied.

*Table 3.* Evaluation metrics when diversity condition holds. Results reported as mean $\pm$ std across 5 random seeds using MLP encoder. Lower APE is better.

| Generative Process | MCC | APE | Final Loss |
|--------------------|-----|-----|------------|
| Identity | $0.775 \pm 0.060$ | $0.007 \pm 0.001$ | $7.388 \pm 0.007$ |
| Linear | $0.854 \pm 0.083$ | $0.007 \pm 0.001$ | $7.386 \pm 0.006$ |
| Invertible MLP | $0.822 \pm 0.038$ | $0.010 \pm 0.002$ | $7.383 \pm 0.012$ |
| Patches | $0.836 \pm 0.060$ | $0.031 \pm 0.001$ | $7.429 \pm 0.019$ |
| Spiral | $0.858 \pm 0.041$ | $0.012 \pm 0.001$ | $7.390 \pm 0.011$ |

### E.2. Diversity Condition Violated

Tables 4-6 compare three approaches when diversity is violated: standard InfoNCE, adapted InfoNCE (Section 4.3), and InfoNCE with inductive bias.

*Table 4.* Mean Correlation Coefficient (MCC) when diversity condition is violated. Results reported as mean $\pm$ std across 5 random seeds.

| Generative Process | Diversity Condition Violated | | |
|--------------------|---------|----------------|-------------------|
| | InfoNCE | InfoNCE Adapted | InfoNCE + Ind. Bias |
| Identity | $0.143 \pm 0.016$ | $0.607 \pm 0.039$ | $0.801 \pm 0.102$ |
| Linear | $0.111 \pm 0.075$ | $0.592 \pm 0.085$ | $0.841 \pm 0.094$ |
| Invertible MLP | $0.178 \pm 0.116$ | $0.569 \pm 0.048$ | N/A |
| Patches | $0.265 \pm 0.018$ | $0.624 \pm 0.035$ | $0.687 \pm 0.014$ |
| Spiral | $0.629 \pm 0.294$ | $0.602 \pm 0.044$ | $0.999 \pm 0.001$ |

*Table 5.* Angular Preservation Error (APE) when diversity condition is violated. Results reported as mean ± std across 5 random seeds. Lower is better.

| Generative Process | Diversity Condition Violated | | |
| --- | --- | --- | --- |
| | **InfoNCE** | **InfoNCE Adapted** | **InfoNCE + Ind. Bias** |
| Identity | $0.322 \pm 0.007$ | $0.152 \pm 0.002$ | $0.047 \pm 0.000$ |
| Linear | $0.325 \pm 0.012$ | $0.155 \pm 0.002$ | $0.046 \pm 0.000$ |
| Invertible MLP | $0.305 \pm 0.034$ | $0.168 \pm 0.003$ | N/A |
| Patches | $0.283 \pm 0.008$ | $0.154 \pm 0.002$ | $0.109 \pm 0.003$ |
| Spiral | $0.136 \pm 0.115$ | $0.156 \pm 0.002$ | $0.016 \pm 0.003$ |

*Table 6.* Final InfoNCE loss when diversity condition is violated. Results reported as mean ± std across 5 random seeds.

| Generative Process | Diversity Condition Violated | | |
| --- | --- | --- | --- |
| | **InfoNCE** | **InfoNCE Adapted** | **InfoNCE + Ind. Bias** |
| Identity | $5.709 \pm 0.000$ | $6.689 \pm 0.007$ | $6.058 \pm 0.009$ |
| Linear | $5.708 \pm 0.001$ | $6.697 \pm 0.005$ | $6.043 \pm 0.009$ |
| Invertible MLP | $5.715 \pm 0.002$ | $6.009 \pm 0.006$ | N/A |
| Patches | $5.850 \pm 0.021$ | $6.028 \pm 0.009$ | $6.222 \pm 0.007$ |
| Spiral | $5.949 \pm 0.140$ | $6.011 \pm 0.012$ | $6.075 \pm 0.014$ |

### E.3. Progressive Diversity Violation

Table 7 and Figure 5c examine how performance degrades as the diversity condition is progressively violated in a 10-dimensional latent space. The parameter $d_{\text{fixed}}$ denotes the number of dimensions held constant during positive pair sampling.

*Table 7.* Linear identifiability ($R^2$) as a function of diversity violation severity for 10D latent space. $d_{\text{fixed}}$ denotes the number of dimensions held constant during positive pair sampling.

| $d_{\text{fixed}}$ | Violation Ratio | **Linear** | **Monomial** |
| --- | --- | --- | --- |
| 0 | 0.0 | $0.969 \pm 0.047$ | $0.990 \pm 0.000$ |
| 1 | 0.1 | $0.023 \pm 0.011$ | $0.015 \pm 0.005$ |
| 2 | 0.2 | $0.014 \pm 0.003$ | $0.015 \pm 0.003$ |
| 3 | 0.3 | $0.027 \pm 0.009$ | $0.034 \pm 0.008$ |
| 4 | 0.4 | $0.061 \pm 0.021$ | $0.042 \pm 0.004$ |
| 5 | 0.5 | $0.099 \pm 0.019$ | $0.097 \pm 0.015$ |
| 6 | 0.6 | $0.168 \pm 0.017$ | $0.185 \pm 0.021$ |
| 7 | 0.7 | $0.213 \pm 0.035$ | $0.284 \pm 0.022$ |
| 8 | 0.8 | $0.238 \pm 0.049$ | $0.348 \pm 0.022$ |
| 9 | 0.9 | $0.299 \pm 0.047$ | $0.425 \pm 0.056$ |
| 10 | 1.0 | $0.032 \pm 0.004$ | $0.086 \pm 0.032$ |

### E.4. Constraint Ratio Experiments

Tables 8-11 show identifiability metrics as a function of the constraint ratio $\rho$, which interpolates between standard InfoNCE ($\rho = 0$) and the fully corrected objective ($\rho = 1$). The parameter $\rho$ controls the fraction of negative samples drawn from the constrained manifold $K(z)$ versus the full sphere $\mathcal{Z}$.

*Table 8.* Linear identifiability ($R^2$) across all generative processes and encoder types as a function of constraint ratio $\rho$. Results reported as mean across 5 seeds.

| $\rho$ | Identity MLP | Identity Inv | Linear MLP | Linear Inv | InvMLP MLP | Patches MLP | Patches Inv | Spiral MLP | Spiral Inv |
|---|---|---|---|---|---|---|---|---|---|
| 0.0 | .053 | .994 | .112 | .995 | .286 | .265 | .871 | .331 | .996 |
| 0.1 | .228 | .997 | .293 | .997 | .317 | .651 | .892 | .731 | .781 |
| 0.2 | .306 | .999 | .299 | .999 | .350 | .763 | .878 | .622 | .963 |
| 0.3 | .226 | 1.00 | .201 | 1.00 | .554 | **.820** | .870 | .797 | .999 |
| 0.4 | .502 | .999 | .452 | .999 | **.666** | .745 | .859 | .813 | .842 |
| 0.5 | .575 | .997 | .487 | .997 | .544 | .737 | .819 | **.822** | .991 |
| 0.6 | .525 | .991 | .461 | .991 | .564 | .719 | .645 | .803 | .831 |
| 0.7 | .469 | .979 | .333 | .979 | .524 | .731 | .674 | .661 | .846 |
| 0.8 | .402 | .958 | .350 | .958 | .485 | .681 | .680 | .620 | .826 |
| 0.9 | .440 | .905 | .457 | .904 | .487 | .675 | .623 | .637 | .811 |
| 1.0 | **.626** | .709 | **.650** | .680 | .597 | .633 | .656 | .624 | .572 |

*Table 9.* Mean Correlation Coefficient (MCC) across all generative processes and encoder types as a function of constraint ratio $\rho$. Results reported as mean across 5 seeds.

| $\rho$ | Identity MLP | Identity Inv | Linear MLP | Linear Inv | InvMLP MLP | Patches MLP | Patches Inv | Spiral MLP | Spiral Inv |
|---|---|---|---|---|---|---|---|---|---|
| 0.0 | .102 | .769 | .125 | .859 | .247 | .275 | .660 | .315 | .997 |
| 0.1 | .342 | .819 | .364 | .836 | .350 | .636 | .648 | .648 | .777 |
| 0.2 | .381 | .815 | .362 | .838 | .453 | .670 | .648 | .632 | .978 |
| 0.3 | .321 | .841 | .336 | .827 | .590 | **.711** | .631 | .685 | .999 |
| 0.4 | .538 | .807 | .542 | .811 | .617 | .664 | .654 | .801 | .876 |
| 0.5 | .594 | .776 | .537 | .834 | .533 | .699 | .628 | **.808** | .995 |
| 0.6 | .522 | .863 | .496 | .818 | .600 | .660 | .540 | .796 | .876 |
| 0.7 | .499 | .778 | .394 | .758 | .561 | .657 | .584 | .628 | .868 |
| 0.8 | .448 | .704 | .408 | .775 | .516 | .649 | .612 | .542 | .861 |
| 0.9 | .463 | .698 | .483 | .703 | .467 | .663 | .560 | .650 | .841 |
| 1.0 | **.607** | .593 | **.592** | .590 | **.569** | .624 | .628 | .602 | .611 |

*Table 10.* Angular Preservation Error (APE) across all generative processes and encoder types as a function of constraint ratio $\rho$. Results reported as mean across 5 seeds. Lower is better.

| $\rho$ | Identity MLP | Identity Inv | Linear MLP | Linear Inv | InvMLP MLP | Patches MLP | Patches Inv | Spiral MLP | Spiral Inv |
|---|---|---|---|---|---|---|---|---|---|
| 0.0 | .325 | .047 | .310 | .046 | .265 | .280 | .112 | .257 | .027 |
| 0.1 | .288 | .035 | .269 | .036 | .265 | .187 | .101 | .134 | .114 |
| 0.2 | .267 | .020 | .268 | .020 | .257 | .146 | .105 | .195 | .038 |
| 0.3 | .286 | **.004** | .293 | **.005** | .210 | **.126** | .107 | .136 | **.011** |
| 0.4 | .223 | .014 | .235 | .014 | .176 | .148 | .111 | .126 | .067 |
| 0.5 | .202 | .034 | .223 | .034 | .204 | .149 | .122 | **.120** | .027 |
| 0.6 | .212 | .054 | .228 | .055 | .197 | .151 | .164 | .123 | .089 |
| 0.7 | .225 | .079 | .255 | .079 | .206 | .145 | .158 | .161 | .088 |
| 0.8 | .238 | .105 | .249 | .106 | .216 | .157 | .152 | .171 | .101 |
| 0.9 | .225 | .138 | .221 | .138 | .212 | .152 | .169 | .165 | .114 |
| 1.0 | **.152** | .155 | **.155** | .158 | **.168** | .154 | .160 | .156 | .190 |

*Table 11.* Final InfoNCE loss across all generative processes and encoder types as a function of constraint ratio $\rho$. Results reported as mean across 5 seeds.

| $\rho$ | Identity MLP | Identity Inv | Linear MLP | Linear Inv | InvMLP MLP | Patches MLP | Patches Inv | Spiral MLP | Spiral Inv |
|---|---|---|---|---|---|---|---|---|---|
| 0.0 | 5.71 | 6.05 | 5.71 | 6.05 | 5.03 | 5.17 | 6.19 | 5.07 | 6.07 |
| 0.1 | 6.07 | 6.18 | 6.06 | 6.18 | 5.38 | 5.50 | 6.31 | 5.45 | 6.44 |
| 0.2 | 6.21 | 6.30 | 6.22 | 6.30 | 5.53 | 5.62 | 6.40 | 5.55 | 6.37 |
| 0.3 | 6.32 | 6.38 | 6.32 | 6.38 | 5.63 | 5.69 | 6.47 | 5.65 | 6.38 |
| 0.4 | 6.40 | 6.46 | 6.40 | 6.46 | 5.71 | 5.76 | 6.53 | 5.72 | 6.53 |
| 0.5 | 6.46 | 6.53 | 6.47 | 6.53 | 5.78 | 5.83 | 6.60 | 5.79 | 6.52 |
| 0.6 | 6.52 | 6.59 | 6.53 | 6.59 | 5.84 | 5.88 | 6.68 | 5.85 | 6.62 |
| 0.7 | 6.57 | 6.62 | 6.57 | 6.63 | 5.89 | 5.93 | 6.71 | 5.89 | 6.72 |
| 0.8 | 6.62 | 6.67 | 6.62 | 6.67 | 5.94 | 5.97 | 6.74 | 5.94 | 6.75 |
| 0.9 | 6.65 | 6.68 | 6.66 | 6.68 | 5.98 | 6.01 | 6.78 | 5.97 | 6.81 |
| 1.0 | 6.69 | 6.69 | 6.70 | 6.69 | 6.01 | 6.03 | 6.79 | 6.01 | 6.94 |

## E.5. Synthetic Validation Figures

Figure 5 provides visual summaries comparing MLP and inverse encoder performance across conditions.

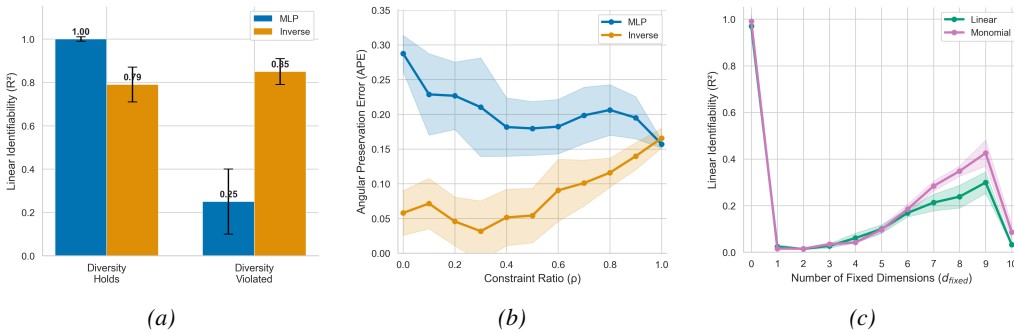

*(a)*        *(b)*        *(c)*

*Figure 5.* Synthetic validation of theoretical predictions. (a) Linear identifiability for MLP and inverse encoders. When the diversity condition holds, MLP achieves $R^2 = 1.00$; when violated, it collapses to $R^2 = 0.25$ while inverse encoders remain robust ($R^2 = 0.83$). (b) Angular preservation error vs. constraint ratio $\rho$. At $\rho = 1$ (corrected InfoNCE), MLP converges to inverse encoder performance. (c) Linear identifiability on $\mathbb{S}^9$ vs. incremental violation of diversity condition; even $d_{\text{fixed}} = 1$ causes catastrophic failure. Results averaged across 5 generative processes; shaded regions indicate $\pm 1$ std.

