# OpenReview forum: "The Loss Is Not Enough: Sampling Conditions and Inductive Bias in Contrastive Representation Learning"
_ICML.cc/2026/Conference — ICML 2026 regular_

### Official Review · Reviewer_nP8n · 2026-02-20

**Soundness:** 2
**Presentation:** 1
**Significance:** 2
**Originality:** 2
**Overall Recommendation:** 2
**Confidence:** 4

**Summary:**

This paper characterizes when InfoNCE-based contrastive learning can recover meaningful latent geometry, which separates two ingredients: (i) a data-sampling condition on how positives are constructed, formalized as a diversity condition, and (ii) model inductive bias that may select geometry-preserving solutions when sampling is insufficient or not fully desirable. Building on the many-negatives (negatives zie $M\to\infty$) cross-entropy perspective of contrastive representation learning attributed to [`1`], the paper argues that under full diversity on a hypersphere representation manifold with vMF positives, minimizers recover the latent space up to orthogonal transforms; but when the diversity condition is violated, the contrastive objective can prefer structure-distorting solutions. Further, the paper proposed a corrected conditional variant that adapts negatives restricted to the positive-pair support.

**Compliance With Llm Reviewing Policy:**

Affirmed.

**Final Justification:**

Given unresolved key concerns about the theoretical scope, the rigor of the key theorem's proof, and the practical relevance of the proposed adapted loss, I would keep my initial score.

**Key Questions For Authors:**

1. What exactly is the new theoretical contribution beyond prior hypersphere identifiability results?

2. Can you specify the reference measure explicitly and state which quantities are densities versus measures, and define symbols at first use?

3. Can you clarify that Theorem 4.4 is about global optima of the objective, and provide a rigorous proof to justify the strict inequality?

4. For correcting the model, how should one implement negatives from $K(z)$ without access to $u$? What realistic proxy approximates the assumption in Eq. (7)?

5. Can you include a real-data experiment (even if the scale is small) for the adapted InfoNCE loss, or explain why it is infeasible?

**Limitations:**

yes

**Strengths And Weaknesses:**

**Strengths:**

1. The paper offers an intuitive framing that loss alone does not guarantee good geometry if the positive-pairing mechanism is support-restricted; proper architecture priors may compensate.

2. The synthetic experiment design cleanly manipulates whether the diversity condition holds versus is violated by freezing a coordinate, and whether the encoder has “more prior” vs a generic MLP. This produces interpretable trends.

3. The adapted InfoNCE objective is a concrete recipe for attempting to link the analytical diagnosis to a methodological intervention.


**Weaknesses:**

1. Theorem 4.3 appears to restate the already-known hypersphere + vMF identifiability result in [`1`]. Also, in Theorem 4.3, assuming the diversity condition and then assuming the positive law subject to vMF is confusing, because full vMF support essentially already implies a strong satisfaction of the diversity condition.

2. Presentation lacks clarity. For example, the integrals mix the using different measure without explanation, e.g., in the diversity condition they use $d z$; in Theorem 3.3 $d \sigma(\cdot)$ while the reference measure is denoted as $\mu$; the shorthand $p(z) \ll p(\tilde z | z)$ mixes the concepts of probability densities and measures; some notations are not introduced or inconsistently used, e.g., interchange $\mathcal{S}$ and $\mathbb{S}$ for denoting a hypersphere, e.g., $k$ and $\sigma$ in the vMF conditional (Eq. (3)) are not defined. Moreover, some formal statements lack clarity, e.g., it’s unclear to readers whether the statement in Theorem 4.4 is about global optima after convergence or about optimization dynamics.

3. The proof of Theorem 4.4 is not rigorous, and appears not to be enough to support the claim. Firstly, the authors treat the *“true conditional”* $p(\tilde z | z)$ as if it were a probability density on the full hypersphere with respect to $d\sigma$, while simultaneously stating it is *“restricted to the submanifold $K(z)$ where $\tilde u = u$”*, which is saying it is singular with respect to the hyperspherical surface measure. In this setting, claims like *“making it impossible for $h$ to minimize the cross-entropy while preserving inner products”* are not well-posed under the reference measure unless the conditional density on $K(z)$ is defined carefully. Secondly, even if we accept the support-mismatch intuition (as stated in the proof *“The support of $q_h(\cdot | z)$ is the entire sphere $\mathcal{Z}$, while the support of $q(\cdot | z)$ is restricted …”*), this point only implies $q_h$ cannot equal $p$ but not justify the much stronger claim *“a non-orthogonal $h$ … can achieve lower loss than any orthogonal transformation”*, nor does it establish the asserted strict inequality $\mathcal{L}(h) < \mathcal{L}(\tilde h), \forall \tilde{h}\in O(k)$.

4. Theorem 4.5 is very close to being by construction. Once the model conditional is constrained to the same support $K(z)$ as the true conditional, showing that orthogonal maps minimize the asymptotic loss is intuitive (as the truth is directly placed back in the model class). Moreover, sampling negatives from $K(z)$ seems to require knowing the invariant component $u$, which is generally not available in real data.

5. Evaluation metric mismatches with the theoretical claims. The theory is about latent recovery and identifiability, but the CIFAR evaluation by linear probe classification accuracy does not directly validate the identifiability-related claim. Moreover, the adapted InfoNCE loss is not convincingly demonstrated on real data.

---
[`1`] Zimmermann, R. S., Sharma, Y., Schneider, S., Bethge, M., & Brendel, W. Contrastive learning inverts the data generating process. In ICML. PMLR, 2021.

---

> ### Author Rebuttal · Authors · 2026-03-31
>
> We thank the reviewer for the thorough review. We address each point below.
>
> **Q1: New theoretical contributions beyond prior identifiability results**
>
> We agree that Theorem 4.3 operates in a similar setting to Zimmermann et al. (2022), but our contributions go substantially beyond in three concrete ways.
>
> First, the **proof machinery is genuinely new**. Zimmermann et al. require differentiability of $h$ to invoke a Jacobian argument (Proposition 2). Our proof applies the *probabilistic Mazur-Ulam theorem* (Zaliaduonis & Gatidis, 2026), which requires no regularity on $h$, yielding a result for a strictly larger class of generative processes; an important distinction, since most real-world generative processes are not differentiable.
>
> Second, we provide a **formal definition of the diversity condition**. Prior work implicitly assumes the sampling mechanism covers the full latent space but never formally defines this requirement. Definition 3.1 is a precise, measure-theoretic characterisation absent from all related work.
>
> Third, we **go substantially beyond the idealised setting**. We prove that violated diversity actively disincentivises orthogonal recovery (Theorem 4.4), propose a corrected objective (Theorem 4.5), and show experimentally that inductive bias is the key compensatory mechanism. None of this has a counterpart in prior work.
>
> **Q1b: Redundancy between diversity condition and vMF**
>
> We agree this was a presentation error. In the revision, Theorem 4.3 will be restated for the vMF setting specifically, with the diversity condition as a *consequence* of vMF rather than an independent hypothesis: (Lemma) vMF implies the diversity condition; (Theorem 4.3) under vMF and uniform marginal, any minimiser of $\mathcal{L}_{CL}$ is an isometry a.e.
>
> **Q2: Notation — reference measures, undefined symbols**
>
> All notation issues will be corrected in the revision. All integrals over the constrained submanifold will use the intrinsic spherical surface measure explicitly, the diversity condition will be rewritten with proper Radon-Nikodym densities, and all undefined symbols will be introduced at first use. The corrected conditional densities are:
>
> $$p(\tilde{z}|z) = \delta(u-\tilde{u}) \cdot \frac{e^{\kappa z^\top \tilde{z}}}{\int_{K(z)} e^{\kappa z^\top z'} d\sigma_{r(z)}^{k-1}(z')}$$
>
> $$q_h(\tilde{z}|z) = \delta(u-\tilde{u}) \cdot \frac{e^{h(z)^\top h(\tilde{z})/\tau}}{\int_{K(z)} e^{h(z)^\top h(z')/\tau} d\sigma_{r(z)}^{k-1}(z')}$$
>
> where $k:=d-\dim(u)$ and $r(z):=\sqrt{1-\|u\|^2}$. The measure $\sigma^{k-1}_{r(z)}$ is essential since the ambient $\sigma^{d-1}$ assigns zero mass to $K(z)$.
>
> **Q3: Theorem 4.4 - scope and rigorous proof**
>
> *Proof:* Define $h_\lambda(u,v):=(u,\lambda v)/\|(u,\lambda v)\|$ for $\lambda>0$, smooth everywhere since $\|(u,\lambda v)\|\geq\|u\|>0$ a.s., making $\lambda=1$ an interior point. Following Wang & Isola (2020), decompose the asymptotic loss into alignment and uniformity terms: $\mathcal{L}(\lambda)=-\frac{1}{\tau}\mathrm{Aln}(\lambda)+\mathrm{Unif}(\lambda)$. Then: (a) all orthogonal maps $A\in O(d)$ achieve the same loss as the identity, since they preserve inner products and $\sigma^{d-1}$; (b) the alignment term satisfies $\mathrm{Aln}(\lambda)>\mathrm{Aln}(1)$ for $\lambda\in(0,1)$, since $f(t):=(\|u\|^2+tv^\top\tilde{v})/(\|u\|^2+t\|v\|^2)$ with $t=\lambda^2$ has $f'(t)\leq 0$ with strict inequality on a positive-measure set; (c) the uniformity term satisfies $\mathrm{Unif}'(1)=0$ (differentiable due to dominated convergence), since by Wang & Isola (2020, Prop. 1) $\sigma^{d-1}$ is the unique minimiser of $\mathrm{Unif}$, making $\lambda=1$ an interior stationary point. As $\mathrm{Aln}(\lambda)-\mathrm{Aln}(1)>0$ dominates $\mathrm{Unif}(\lambda)-\mathrm{Unif}(1)=o(1-\lambda)$, taking $h^*:=h_\lambda$ for $\lambda$ close to 1 completes the proof.
>
> **Q3b: Theorem 4.5**
>
> We agree the result is by design: matching the support of the model conditional to the true conditional places orthogonal solutions back in the minimiser set. Non-uniqueness is precisely why inductive bias remains necessary to select among numerically equivalent solutions.
>
> **Q4: Implementing negatives from $K(z)$**
>
> In synthetic settings with known augmentation structure, negatives can be drawn from the same augmentation orbit. In general practice $K(z)$ is inaccessible since $u$ is unknown, but can be approximated by applying the same data augmentations as for the positive samples. That's why practically the inductive bias is essential.
>
> **Q5: CIFAR metrics and adapted loss at scale**
>
> CIFAR-10 validates two predictions: the gap between augmentation regimes confirms the diversity condition matters in practice; the widening gap between architectures confirms the compensatory role of inductive bias (Theorem 4.4). The adapted loss is not evaluated on real data due to its $O(N\cdot M)$ memory cost versus $O(N)$ for standard InfoNCE; memory-efficient approximations are left as future work (Section 7).

---

> > ### Author Rebuttal · Reviewer_nP8n · 2026-04-02
> >
> > Thank you for your response. While several concerns were resolved, some of my major concerns persist.
> >
> > First, the use of the probabilistic Mazur-Ulam theorem is indeed a new proof technique; however, the rebuttal does not make it clear which practically relevant generative processes fall in the new but not the old setting. It remains unclear how much the theoretical scope has expanded, or whether new identifiability insights have been provided.
> >
> > Moreover, the proof of Theorem 4.4 still lacks rigor. It glosses over the conditions necessary for the uniformity term to dominate, as well as the interpolation argument. Additionally, justifying the strict loss inequality (the central claim) requires more than just the support mismatch, which is entirely not discussed.
> >
> > Finally, the author acknowledged that the adapted InfoNCE loss remains entirely unevaluated on real data, with the reason that the memory cost $O(NM)$. While I understand the computational constraint, this largely undermines the practical value of the proposed adapted learning objective if it is entirely inaccessible at the scale of even a modest dataset like CIFAR-10.
> >
> > Therefore, I would keep my score at 2.

---

> > > ### Author Response · Authors · 2026-04-07
> > >
> > > Thank you for your comments.
> > >
> > > Regarding the rigor of the proof - we apologize for succinctness, as we only had 5000 characters. Here is the justification for why the alignment gain dominates locally.
> > >
> > > **Alignment gain is first-order.** Since positive pairs share the same $u$ and $\|u\|^2 + \|v\|^2 = 1$, differentiating $h_\lambda(z)^\top h_\lambda(\tilde{z})$ with respect to $\lambda$ at $\lambda = 1$ gives a strictly positive quantity on a set of positive measure (as $\|u\| > 0$ and $v \neq \tilde{v}$), so $\mathrm{Aln}'(1) > 0$ and $\mathrm{Aln}(\lambda) - \mathrm{Aln}(1) = \Theta(1-\lambda)$.
> > >
> > > **Uniformity cost is sub-first-order.** At $\lambda = 1$, $h_1 = I$ pushes forward $\sigma^{d-1}$ to itself, which is the unique minimizer of $\mathrm{Unif}$ (Wang & Isola, 2020, Proposition 1). Hence $\mathrm{Unif}'(1) = 0$, and by Taylor expansion $\mathrm{Unif}(\lambda) - \mathrm{Unif}(1) = o(1-\lambda)$.
> > >
> > > **Conclusion.**
> > >
> > > $$\mathcal{L}(\lambda) - \mathcal{L}(1) = -\frac{1}{\tau}\Theta(1-\lambda) + o(1-\lambda) < 0$$
> > >
> > > for $\lambda$ sufficiently close to $1$ from below.
> > >
> > > **On the scope of the theoretical contribution.** We would like to emphasize that the adapted loss is primarily a theoretical vehicle: it demonstrates that even when the underlying sampling model is fully known and the loss is adapted to it, the original feature space cannot be recovered. This gives precise insight into when and why violation of the diversity condition is fundamental, not an artifact of a suboptimal loss. The probabilistic Mazur-Ulam theorem extends this to a strictly broader class of generative processes than prior work, specifically processes that are neither continuous nor differentiable. A concrete and practically relevant example is the Patches generative process (Section 5.1, Figure 2d), which applies piecewise rotations creating discontinuities, making it neither continuous nor differentiable. In practice, the true generative process is unknown, so we cannot formally verify which class it belongs to; however, it is safe to assume that most real-world generative processes are not differentiable, making the expanded theoretical scope practically meaningful. We agree that making this explicit in the paper is valuable and will clarify this in the revision.
> > >
> > > **On the computational cost of the adapted loss.** We would like to strongly clarify that the adapted InfoNCE loss is not proposed as a practical solution. It is a theoretical argument: even if the sampling model were fully known and the loss were adapted to it, the diversity condition violation cannot be compensated for by adjusting the sampling mechanism alone. This is the central insight. What follows from it is that exploiting the inductive bias of the encoder is necessary, and this is precisely what we demonstrate empirically with synthetic experiments. The computational intractability of the adapted loss does not undermine the theoretical conclusion; it simply confirms that the fix must come from the encoder side, not the loss side.

---

### Official Review · Reviewer_ibvW · 2026-03-10

**Soundness:** 2
**Presentation:** 1
**Significance:** 2
**Originality:** 4
**Overall Recommendation:** 3
**Confidence:** 3

**Summary:**

This paper provides a theoretical characterization of the conditions under which a contrastive learning–based encoder learns linearly separable latent representations. It also introduces a generalized InfoNCE loss designed to improve performance when these conditions do not hold.

**Compliance With Llm Reviewing Policy:**

Affirmed.

**Final Justification:**

Overall, I recommend rejection.

I did not receive any follow-ups from the authors regarding my concerns about the clarification of several important concepts, e.g., “appearance-only” or “structure-preserving” augmentation.

**Key Questions For Authors:**

Could you compare with Toward Understanding the Feature Learning Process of Self-supervised Contrastive Learning?

**Limitations:**

yes

**Strengths And Weaknesses:**

**Strengths**

1. The paper formalizes the diversity condition for latent space sampling and provides proofs for the necessary conditions for isometric recovery.

2. It proposes a generalized InfoNCE objective to enable isometric embedding, which is important for effective representation learning.

3. The paper includes extensive experiments on synthetic data to support and illustrate the theoretical insights.


**Weaknesses**

1. The presentation requires significant improvement. For example, it is unclear what the exact form of the proposed loss function is. Additionally, the statement *“this can be realized by generating negatives through transformations that preserve u while perturbing v”* is difficult to interpret. More explanation is needed to clarify how this mechanism works in practice. Similarly, the conditions and results of the theorems are hard to interpret without clearer connections to practical settings.

2. The paper does not sufficiently discuss existing theoretical analyses of contrastive learning.

3. The sections on assumptions and proof sketch are needed.

---

> ### Author Rebuttal · Authors · 2026-03-31
>
> We thank the reviewer for the constructive feedback. We address each point below.
>
> **Q1: The exact form of the proposed loss is not clearly stated**
>
> We agree the loss formulation warrants a clearer presentation. The adapted InfoNCE loss differs from the classical one in a single way: negatives are drawn uniformly from the submanifold $K(z)$ determined by the anchor $z$, rather than from the full latent space. Given positive pair $(z, \tilde{z}) \sim p_{\text{pos}}$ and $M$ negatives $z^-_1, \ldots, z^-_M \sim U(K(z))$, the per-sample loss is:
>
> $$\ell = -\log\frac{\exp(h(z)^\top h(\tilde{z})/\tau)}{\exp(h(z)^\top h(\tilde{z})/\tau)+\sum_{j=1}^{M}\exp(h(z^-_j)^\top h(z)/\tau)}$$
>
> The full adapted InfoNCE loss is the expectation of the above over the joint distribution of positive pairs and negatives (see Definition 3.2 in the paper for the complete expression with sampling notation).
>
> In practice, $K(z)$ is approximated by applying the same augmentation $T$ independently $M$ times to the same anchor $x_i$, rather than using other batch elements as negatives. Since $T$ fixes semantic content $u$ while perturbing style $v$, all $M$ negatives share the same $u_i$, approximating $U(K(z_i))$. A full pseudocode implementation is provided in the revision of the reviewer Ft7r.
>
> **Q2: "Generating negatives through transformations that preserve u while perturbing v" is hard to interpret**
>
> Recall that $z = (u, v)$ decomposes into a non-varying component $u$ (the latent dimensions held fixed under the conditional $p(\tilde{z}|z)$) and a varying component $v$ (the dimensions that are perturbed). An augmentation that "preserves $u$ while perturbing $v$" is one that leaves the fixed latent dimensions unchanged while changing the varying ones.
>
> **Concrete example.** Consider an anchor image of a cat, where $u$ encodes its identity and spatial structure, and $v$ encodes its appearance (brightness, color, texture). In standard InfoNCE, negatives are other batch images (other cats, dogs, cars), which differ in both $u$ and $v$, creating the support mismatch described in Section 4.2. In the adapted loss, we instead apply appearance-only augmentations (color jitter, blur, brightness) independently $M$ times to the same image. All $M$ negatives then have the same $u$ (same cat, same spatial structure) with different $v$ (different appearance), approximating a uniform draw from $K(z)$.
>
> **Q3: Theorem conditions are hard to connect to practical settings**
>
> Our model assumes a spherical latent space underlying a higher-dimensional data manifold. Contrastive learning reconstructs this space because natural co-occurring pairs approximately follow a vMF conditional density: more frequent co-occurrence corresponds to closer latent proximity. This is consistent with the empirical finding in SimCLR (Chen et al., 2020) that not all augmentations yield equally good representations; we connect this to how well each augmentation regime approximates the diversity condition. Augmentations that perturb more latent dimensions yield more expressive representations, as shown in Figure 3 of our paper. The adapted loss provides a principled remedy when diversity is violated.
>
> **Q4: Comparison with Wen & Li (2021)**
>
> We thank the reviewer for this pointer and will add Wen & Li (2021) to related work. The two works are fundamentally different. (1) *Regime:* they analyse SGD dynamics on finite networks; we analyze the asymptotic cross-entropy objective ($M\to\infty$). (2) *Question:* they ask which features gradient descent learns; we ask whether the representation recovers latent geometry up to isometry. (3) *Inductive bias:* Wen & Li show augmentations are necessary for good features, but do not analyse the compensatory role of architectural inductive bias when diversity is violated. Characterizing precisely when the objective fails, and showing that inductive bias compensates for the violation.
>
> **Q5: Dedicated section on assumptions and proof sketches**
>
> We agree and will add this in the revision. The consolidated assumptions (to be collected into a single block before the theorems) are:
>
> - **A1 (Latent space):** $Z = S^{d-1}$, the unit hypersphere in $\mathbb{R}^d$
> - **A2 (Marginal):** $P_Z$ is the uniform distribution on $S^{d-1}$
> - **A3 (Conditional):** $P_{\tilde{Z}|z}$ follows a von Mises-Fisher distribution with concentration $\kappa>0$
> - **A4 (Encoder capacity):** The encoder $h$ has infinite capacity
> - **A5 (Normalization):** Representations are $L^2$-normalised, i.e. $h:\mathcal{X}\to S^{d-1}$
>
> We will add inline proof sketches for each theorem. Theorem 4.3 proceeds by showing the loss converges to a KL divergence (using A1-A4), forcing conditional matching and inner product preservation, then by probabilistic Mazur-Ulam theorem to obtain a global orthogonal map. Theorem 4.4 constructs an explicit witness $h_\lambda$ exploiting the support mismatch caused by A5. Theorem 4.5 shows that correcting the support restores orthogonal minimizers.

---

> > ### Author Rebuttal · Reviewer_ibvW · 2026-04-02
> >
> > Thank you for the rebuttal. While the concrete example is helpful, my main concern remains unresolved. The notion of applying augmentations independently is not clearly justified: in what sense are two augmentations assumed to be independent, and what principle supports this assumption? In addition, the paper does not provide a formal definition of “appearance-only” or “structure-preserving” augmentation. These notions seem task-dependent, since whether a transformation preserves relevant information depends on the downstream objective. If so, such distinctions cannot be meaningfully determined at the pre-training stage.

---

> > > ### Author Response · Authors · 2026-04-07
> > >
> > > **On augmentation independence and structure-preserving augmentations.** We would like to clarify several points.
> > >
> > > Augmentations are just one instance of a sampling mechanism in our theoretical setting. The framework applies equally to other mechanisms such as sampling temporally close frames in video, or spatially close patches from an image. The independence assumption is not specific to augmentations; it reflects the standard statistical assumption that positive pairs are drawn from the conditional distribution $p(\tilde{z}|z)$, which is defined independently per anchor.
> > >
> > > Our analysis does not make any value distinction between generative factors, nor does it depend on any downstream task. The examples of "appearance" vs. "structure" were purely illustrative of how different dimensions of the generative process might behave in practice. We do not formally define these notions, nor do we need to: our theoretical conclusion is task-agnostic. Specifically, we show that in order to obtain representations that are useful for almost all downstream tasks, the sampling mechanism must vary samples across all dimensions of the generative process, which is precisely the diversity condition.
> > >
> > > Our core insight is that no dimension should remain constant. Cropping is a practical example that approximately satisfies the diversity condition because it varies almost all generative factors simultaneously (geometry, color, and local statistics). By contrast, augmentations such as rotation or color jitter vary only a small subset of dimensions, heavily violating the diversity condition. We do not prescribe which factors are semantically meaningful; we only observe that keeping any dimension fixed is theoretically harmful, which reflects the practical setting, and inductive bias is the property that makes contrastive learning work.

---

### Official Review · Reviewer_Ft7r · 2026-03-13

**Soundness:** 3
**Presentation:** 3
**Significance:** 3
**Originality:** 3
**Overall Recommendation:** 5
**Confidence:** 3

**Summary:**

This paper proposes a theoretical analysis of the conditions under which contrastive learning can recover meaningful latent representations for downstream tasks. They introduce a diversity condition on the conditional distribution of positive samples and show that it is necessary for reconstructing the latent space when the encoder is sufficiently expressive. They prove that when this condition holds, CL can recover the latent structure up to an orthogonal transformation. Through synthetic experiments, they show that an MLP encoder achieves near-perfect linear identifiability when diversity holds, while violating the diversity condition leads to collapsed and poorly structured representations.

**Compliance With Llm Reviewing Policy:**

Affirmed.

**Final Justification:**

Overall, while I still find the paper interesting and potentially valuable, the rebuttal has not fully addressed my concerns regarding generality and empirical validation. As a result, I am now less certain about the strength of my initial recommendation.

**Key Questions For Authors:**

1. What happens to the results of theorem 4.3 and 4.4 if the assumptions (i.e. conditional follows a von MIses-Fisher distribution, and constraining the latent space to a spherical manifold) are violated?

2. The proposed solution for InfoNCE draw negative samples from the same manifold K(z). In practice, however, it is not clear how this would be implemented. Do you have any practical algorithm that approximate your approach?

3. You mention that inductive bias can help when the diversity condition is violated. What kind of inductive biases or architectural biases are effective? am particularly interested because inductive biases can also be harmful in contrastive learning in other scenarios—for example, when spurious correlations are present in the data—so it would be useful to understand when such biases are beneficial versus detrimental.

**Limitations:**

Yes.

**Strengths And Weaknesses:**

Strengths:
1. To my knowledge, the paper provides a clear theoretical contribution that the diversity condition is necessary for reconstructing the latent space in CL.
2. The theoretical analysis is supported by controlled synthetic experiments.

Weaknesses:
1. The paper doesn't provide a fully general solution to the problem. They propose sampling negatives from the same manifold K(z) instead of the full space, this modification does not completely resolve the issue. As the authors acknowledge, the solution is not unique and inductive bias in the encoder is still required to recover the latent geometry.
2. The experiments are limited to cifar10. Validation on more complex datasets would strengthen the significance of the theoretical results.

---

> ### Author Rebuttal · Authors · 2026-03-31
>
> We thank the reviewer for the positive assessment and constructive feedback.
>
> **Weakness 1: The paper does not provide a fully general solution**
>
> We want to be precise about what the paper claims. It should be understood as an analysis of the complementary roles of the **sampling mechanism** and **inductive bias** in contrastive learning framework. Our results give a precise answer to what each component can achieve alone and how they interact:
>
> - **Standard InfoNCE under violated diversity (Theorem 4.4):** The loss actively disincentivizes orthogonal recovery. No encoder architecture can overcome this.
> - **Adapted InfoNCE (Theorem 4.5):** Orthogonal recovery becomes achievable - it enters the set of global minimizers. This shows the support mismatch was the precise obstacle.
> - **Why inductive bias is still necessary:** The adapted loss has non-unique minimizers (Theorem A.5). Inductive bias selects among them in favor of geometry-preserving solutions.
>
> The sampling mechanism determines which solutions are achievable; inductive bias determines which is found. Neither alone is sufficient. We will make this framing explicit in the revision.
>
> **Weakness 2: Experiments limited to CIFAR-10**
>
> The primary constraint is computational: the adapted InfoNCE requires $O(N \cdot M)$ memory, effectively $O(N^2)$ for a fair comparison setting, and contrastive learning requires large batches, making larger datasets infeasible under our compute budget.
>
> **Q1: What happens to Theorems 4.3 and 4.4 if vMF and spherical assumptions are violated?**
>
> Extending to more general geometries (beyond the applications where cosine similarity is the primary metric of semantic difference) and different classes of conditional densities is a direction for future work. We adopt the spherical latent space and vMF conditional because vMF intuitively captures co-occurrence density in natural data - nearby latent points appear as positive pairs more frequently, and provides a clean parametrization via $\kappa$, making the framework mathematically tractable and empirically motivated.
>
> **Q2: Is there a practical algorithm that approximates sampling negatives from $K(z)$?**
>
> Thank you for a very relevant question, we will provide the exact implementation in the appendix.
>
> ```
> Input: encoder h, batch {x_1,...,x_N}, augmentation T, temperature τ, M negatives
>
> For each anchor x_i:
>   1. POSITIVE:  x̃_i  ← T(x_i)          (one stochastic draw)
>   2. NEGATIVES: x⁻_{i,j} ← T(x_i)      (M independent draws, j=1..M)
>   3. LOSS:
>        L_i = -log [ exp(h(x_i)ᵀh(x̃_i)/τ) /
>                     (exp(h(x_i)ᵀh(x̃_i)/τ) + Σⱼ exp(h(x_i)ᵀh(x⁻_{i,j})/τ)) ]
>
> Return: L = (1/N) Σᵢ Lᵢ
> ```
>
> The sole difference from standard InfoNCE is that negatives come from repeated augmentation of the **same anchor** $x_i$, rather than from other batch elements (this way sampling from approximately the same manifold). Since $T$ fixes the invariable component $u$ while perturbing the variable component $v$, all $M+1$ samples share the same $u_i$, approximating $U(S^{k-1}_{r(z_i)})$. This approximation samples $v$ from the augmentation distribution rather than uniformly from $K(z)$, which we note as a limitation.
>
> **Q3: What kinds of inductive biases are effective?**
>
> An effective inductive bias should: (1) parameterize an approximate inverse of $g$ - e.g. a CNN for image data where $g$ involves spatial composition of local features; and (2) have as small a parameter space as possible, preventing the optimizer from finding non-geometry-preserving minimizers that satisfy the loss (Theorem A.5). An MLP satisfies neither condition for natural image data.

---

> > ### Author Rebuttal · Reviewer_Ft7r · 2026-04-06
> >
> > Thank you for the rebuttal.
> > Regarding Weakness 1, the distinction that sampling determines the set of achievable solutions while inductive bias selects among them is a valuable insight and improves the framing of the paper. However, my concern remains partially unresolved: while the adapted InfoNCE removes the support mismatch and makes orthogonal recovery achievable, it does not guarantee it, and still relies on inductive bias. As a result, the proposed modification does not constitute a fully general solution, but rather shifts part of the burden to the model design. I encourage the authors to emphasize this limitation more explicitly.
> > For Q2, the proposed approximation is interesting, but sampling negatives from repeated augmentations of the same anchor changes the nature of the contrastive task (e.g., reducing instance discrimination across samples). It would be useful to discuss more explicitly how this affects representation quality and what trade-offs arise compared to standard negative sampling.
> > For Q3, the explanation of inductive bias is insightful, particularly the connection to approximating the inverse of the generative process and restricting the hypothesis space. However, it would be valuable to also discuss potential failure cases, especially when inductive biases reinforce spurious correlations, which can be problematic in contrastive learning.

---

> > > ### Author Response · Authors · 2026-04-07
> > >
> > > **Regarding Weakness 1.** We agree with the reviewer that the adapted InfoNCE does not constitute a fully general solution: it removes the support mismatch and makes orthogonal recovery achievable, but does not guarantee it, and inductive bias remains necessary to select geometry-preserving solutions among the minimizers. We agree that developing more memory-efficient approximations of this sampling strategy is an important direction for future work to scale this method to more practical settings, and we have included it explicitly as such in Section 7.
> > >
> > > **Regarding Q2.** Contrastive learning, in our theoretical framing, is not fundamentally about instance discrimination across samples. Rather, the learned representations capture distance structure in the latent space: since the conditional distribution $p(\tilde{z}|z)$ approximates a vMF distribution, sampling frequency is inversely related to latent distance from the anchor. The contrastive objective therefore has access to latent distance information encoded in co-occurrence frequency, and the goal is to recover this distance structure, not to discriminate between specific instances. Under this view, drawing negatives from repeated augmentations of the same anchor is consistent with the theoretical objective: it constrains negatives to the same submanifold $K(z)$, correcting the support mismatch and making orthogonal recovery achievable (Theorem 4.5). The representation quality achieved by this approach is validated quantitatively in Table 1 (InfoNCE Adapted), where it substantially improves over standard InfoNCE under violated diversity, and confirmed by the angular preservation error results. We agree that developing more memory-efficient approximations of this sampling strategy is an important direction for future work, and we have included it explicitly as such in Section 7.
> > >
> > > **Regarding Q3.** We agree this is a valid and important failure case. In our framework, inductive bias works precisely when it approximates the inverse of the true generative process $g^{-1}$. When the bias is mismatched with $g$, the encoder recovers the wrong structure, and the specific spurious correlations that emerge depend on what the architecture is sensitive to. Crucially, if the augmentation pipeline consistently preserves these spurious features across positive pairs, the contrastive objective will actively reinforce them: if the spurious correlations happen to reduce the loss, the optimizer will exploit them, potentially leading to worse downstream representations than a less biased encoder would produce. We note that this failure mode is not unique to our framework; it applies to all methods that rely on inductive bias to constrain the hypothesis space. Our contribution is to make it explicit: the degree of failure is governed by the mismatch between the inductive bias and $g$, which provides a principled basis for diagnosing and potentially correcting it. We will add this discussion in the revision.

---

### Official Review · Reviewer_4U16 · 2026-03-17

**Soundness:** 4
**Presentation:** 4
**Significance:** 3
**Originality:** 3
**Overall Recommendation:** 5
**Confidence:** 3

**Summary:**

They use measure theory to define the necessary conditions for latent space construction for contrastive learning. They provide a necessary diversity condition that allow for linear identifiability and recovery of the latent space. In this regard, they propose a contrastive loss to account for violation of the diversity condition. They verify their proposed condition and that when that is satisfied, the infoNCE successfully recover the latent rep. They show that when the diversity condition is violated, an inductive bias of the encoder becomes important; a correct inductive bias can with InfoNCE will enable the recovery. Moreover, they propose a modified contrastive bias with in the presence of violated diversity condition it can improve the recovery as opposed to the original InfoNCE.

**Compliance With Llm Reviewing Policy:**

Affirmed.

**Final Justification:**

Here is my final comment from the rebuttal.

Majority of my comments related to increasing the clarity of the paper. The authors have addressed them in this rebuttal. Moreover, the paper in my opinion has enough contributions (even that it's building on prior work Zimmermann et al. (2022) for theory) to meet the bar for publication.

**Key Questions For Authors:**

See above.

**Limitations:**

See above.

**Strengths And Weaknesses:**

Strengths are:

- The paper is written clearly and organized well. I enjoyed reading the paper. I found the literature review and points to related works informative and have proper scope to give the reader sufficient information to be able to understand the position of the paper with respect to the literature.

- The paper provide excellent intuition in numerous part of the paper for the reader to be able to understand the implications of the technical content (e.g., line 260, section 3.5, section 4.3, ...).

- I find the paper to provide detailed information to be able to reproduce the results and experiments.

- Regarding the topic, this is a timely paper which discusses identifiability theory of latent representation with contrastive learning. It provides rigorous formulation, results, backed by experiments. The community of representation learning would find the paper interesting and useful.

----

I did not find any particular weakness. The paper is through and through my first read, I found the mathematical formulation proper.


I only have a few questions and suggestions on clarity:

- For general reader, please introduce what InfoNCE is when first used in the text.

- I read the definition on diversity several times; could the author provide more elaborations on how they come with such definition and condition more intuitively on page 3.

- Could the authors explain prior to Section 5 what would be the inductive bias on the encoder for geometry-preserving solutions?

- How did you evaluate the inductive bias the network for CIFAR experiment and rate them as low/high for the geometry distance preserving mapping?

---

> ### Author Rebuttal · Authors · 2026-03-31
>
> We thank the reviewer for their careful reading and constructive feedback. We address each point below.
>
> **Q1: Please introduce InfoNCE when first used in the text**
>
> Thank you for this suggestion. We will add a brief introduction at first use. InfoNCE (Information Noise Contrastive Estimation) was introduced by van den Oord et al. (2019). We will add this introduction before Definition 3.2 in the revision.
>
> **Q2: Could you provide more intuition on how the diversity condition was derived?**
>
> The diversity condition was not derived from first principles. Rather, we posed the question of what would happen if we followed a realistic sampling pattern and restricted the conditional density to a smaller support that models augmentation procedures used in practice.
>
> The starting point is the theoretical ideal used by Zimmermann et al. (2022): the vMF conditional has full support on $S^{d-1}$, meaning positive pairs can cover the entire latent space for any anchor $z$. This allows full latent space recovery. However, this idealized setting contradicts practical data augmentation: full-support conditionals would in principle allow augmenting one data class into another. Real augmentations restrict the conditional to some smaller region of the latent space, not the full sphere.
>
> This observation motivated us to ask: what is the weakest condition on the conditional that still guarantees latent space recovery? The answer is the diversity condition $P_Z \ll P_{\tilde{Z}|z}$: the conditional measure just needs to be non-zero wherever the marginal $P_Z$ is non-zero. Intuitively, if we can observe almost all pairs from the latent space, we can estimate all pairwise distances, and therefore reconstruct it. Absolute continuity is the minimal mathematical condition encoding this guarantee.
>
> **Q3: Could you explain what inductive bias on the encoder means for geometry-preserving solutions, prior to Section 5?**
>
> Yes, we will incorporate this into the paper. In our framework, an encoder $h$ is geometry-preserving if it recovers the latent structure up to an orthogonal transformation, i.e. $h \approx A \circ g^{-1}$ for some $A \in O(d)$. When the diversity condition is violated, the contrastive loss alone cannot enforce this; there are many global minimizers, most of which are not geometry-preserving (Theorem A.5). Inductive bias on the encoder restricts the hypothesis space to maps that approximate $g^{-1}$, making geometry-preserving solutions the default rather than a special case.
>
> Concretely, an effective inductive bias should satisfy two conditions: (1) it should parameterize an approximate inverse of the data-generating process $g$ — for example, a CNN for image data where $g$ involves spatial composition of local features; and (2) it should have as small a parameter space as possible, so that the optimizer cannot find non-geometry-preserving minimizers that satisfy the loss. An MLP, despite being a universal approximator, satisfies neither condition for natural image data, and therefore performs poorly under violated diversity.
>
> **Q4: How was inductive bias evaluated and rated as low/high for the CIFAR architectures?**
>
> The rating reflects how well each architecture's structural priors align with the spatial generative structure of natural images, i.e. how closely the architecture parameterizes an inverse of the image-generating process $g$.
>
> - **ResNet-18 (high inductive bias):** Convolutional layers encode spatial locality and translation equivariance, directly mirroring the spatial composition of local features in natural images. This gives ResNet-18 a strong prior toward geometry-preserving solutions when spatial structure is a true latent factor.
> - **ViT (medium inductive bias):** Self-attention operates globally across patches without hard spatial priors. It can in principle learn spatial structure, but does not encode it by construction, leaving more room for non-geometry-preserving solutions.
> - **MLP (low inductive bias):** Treats the image as a flat vector with no structural assumptions. As a universal approximator with no domain-specific constraints, it imposes virtually no restriction on the hypothesis space and therefore provides the weakest geometry-preserving prior.
>
> We will make this rating criterion explicit in the revision alongside Table 1.

---

> > ### Author Rebuttal · Reviewer_4U16 · 2026-04-02
> >
> > I thank the authors for the rebuttal response. Majority of my comments related to increasing the clarity of the paper.  The authors have addressed them in this rebuttal. Moreover, the paper in my opinion has enough contributions (even that it's building on prior work Zimmermann et al. (2022) for theory) to meet the bar for publication.

---

### Decision · Program_Chairs · 2026-04-30

**Decision:**

Accept (regular)

**Comment:**

The paper uses measure theory to define the necessary conditions for latent space construction for contrastive learning.

The paper presents excellent intuition to understand the implications of the work, and presents theory of latent representations with contrastive learning.

The paper doesn't sufficiently discuss other theoretical analysis of contrastive learning (not providing a fully general solution), and the experiments are limited.

Reviewer 4U16 recommended an accept after the rebuttal mentioning that most of the concerns were resolved during the rebuttal and that has enough contributions.

Reviewer Ft7r recommended an accept after the rebuttal even if it didn't fully addressed all their concerns.

Reviewer ibvW recommended a weak reject after the rebuttal claiming not to have received an answer to their concerns about the clarification of several important concepts.  The authors replied after the reviewer's final assessment.

Reviewer nP8n recommended a reject after the rebuttal given the unresolved key concerns about the theoretical proof and practical relevance.

Overall, the paper received mixed reviews.  The reviewers that suggest acceptance are enthusiastic about paper and its contributions despite their flaws.  From the negative reviewers, one has not significant concerns, while the latter has valid concerns regarding theoretical proof.  While the authors claim to have solved the issues during the rebuttal.  Given the strengths of the proposal, I lean towards recommending its acceptance.